# Molecular mechanism of biased signaling at the kappa opioid receptor

Amal El Daibani[1,2,11], Joseph M. Paggi[3,11], Kuglae Kim[4,10,11], Yianni D. Laloudakis[3,11], Petr Popov[5], Sarah M. Bernhard [1,2], Brian E. Krumm [4], Reid H. J. Olsen[4], Jeffrey Diberto[4], F. Ivy Carroll[6], Vsevolod Katritch [7], Bernhard Wünsch [8], Ron O. Dror [3,9] ✉ & Tao Che [1,2] ✉

The κ-opioid receptor (KOR) has emerged as an attractive drug target for pain management without addiction, and biased signaling through particular pathways of KOR may be key to maintaining this benefit while minimizing side-effect liabilities. As for most G protein-coupled receptors (GPCRs), however, the molecular mechanisms of ligand-specific signaling at KOR have remained unclear. To better understand the molecular determinants of KOR signaling bias, we apply structure determination, atomic-level molecular dynamics (MD) simulations, and functional assays. We determine a crystal structure of KOR bound to the G protein-biased agonist nalfurafine, the first approved KOR-targeting drug. We also identify an arrestin-biased KOR agonist, WMS-X600. Using MD simulations of KOR bound to nalfurafine, WMS-X600, and a balanced agonist U50,488, we identify three active-state receptor conformations, including one that appears to favor arrestin signaling over G protein signaling and another that appears to favor G protein signaling over arrestin signaling. These results, combined with mutagenesis validation, provide a molecular explanation of how agonists achieve biased signaling at KOR.

G protein-coupled receptors (GPCRs) are membrane proteins that can bind to numerous ligands and transmit signals into cells. In response to binding of extracellular ligands, GPCRs activate intracellular signal transducers, such as G proteins, arrestins, and/or GPCR kinases. The activation of these signal transducers can then trigger non-overlapping signaling pathways that determine the ligand-specific responses[1–3]. Much attention has been focused on biased signaling—a phenomenon in which certain ligands preferentially activate one signaling pathway over the other—as a potential strategy to preserve therapeutic benefits without unwanted side effects. Although several biased ligands have

been identified in preclinical and clinical studies that show reduced side-effect liabilities[4–7], the molecular mechanisms by which different agonists selectively stimulate different signaling pathways have remained unclear. It is widely believed that signaling bias is a common phenomenon as dynamic conformational changes induced by ligand binding could lead to preferential coupling of specific transducers[8,9]. While many structures of GPCRs bound to G proteins or arrestins have been produced in attempts to elucidate the structural or molecular features responsible for signaling bias[1], there have been no identifiable generalized rules as the differences between these structures are

[1]Department of Anesthesiology, Washington University School of Medicine, Saint Louis, MO, USA. [2]Center for Clinical Pharmacology, University of Health Sciences & Pharmacy and Washington University School of Medicine, Saint Louis, MO, USA. [3]Department of Computer Science, Stanford University, Stanford, CA, USA. [4]Department of Pharmacology, University of North Carolina School of Medicine, Chapel Hill, NC, USA. [5]iMolecule, Skolkovo Institute of Science and Technology, Moscow, Russia. [6]Research Triangle Institute, P.O. Box 12194, Research Triangle Park, NC 27709, USA. [7]Department of Biological Sciences, University of Southern California, Los Angeles, CA, USA. [8]Institut für Pharmazeutische und Medizinische Chemie, Universität Münster, Corrensstraße 48, 48149 Münster, Germany. [9]Departments of Molecular and Cellular Physiology and of Structural Biology, Stanford University School of Medicine, Stanford, CA, USA. [10]Present address: Department of Pharmacy, Yonsei University, Incheon 21983, Republic of Korea. [11]These authors contributed equally: Amal El Daibani, Joseph M. Paggi, Kuglae Kim, Yianni D. Laloudakis. ✉e-mail: ron.dror@stanford.edu; taoche@wustl.edu

subtle. However, nuclear magnetic resonance (NMR)[10], double electron-electron resonance (DEER)[11], and molecular dynamics (MD) simulations[12] approaches have indicated that ligand-specific transducer couplings appear to be more related to dynamic conformational changes in the intracellular regions of the receptor, such as transmembrane domain (TM) 7 and helix 8.

The κ-opioid receptor (KOR) plays a unique role among the opioid receptor family as its activation antagonizes the activity of other opioid receptors in several aspects. Opioid receptors, particularly the μ-opioid receptor (MOR), are the primary targets for opiate drugs to alleviate pain. However, opioid abuse, addiction, and overdose are at epidemic proportions in the US[13]. The magnitude of these problems has led to a search for opioid alternatives to treat pain and related conditions. As KOR agonists have a low associated risk of addiction and minimal gastrointestinal and respiratory side effects, their use in the development of new pain therapies is promising. However, dysphoria and hallucination effects have limited their clinical utility in humans[14–18]. Biased selective agonists for KOR that trigger receptor activity toward the G-protein pathways rather than arrestin have been the subject of interest for their analgesic effect and as a safer alternative to opioids with reduced undesirable effects[19–22]. For example, nalfurafine is the first clinically approved KOR agonist to treat uremic and chronic pruritus in hemodialysis patients without producing dysphoria and hallucination at therapeutic doses[23,24]. Nalfurafine has been reported to be a G protein-biased agonist that displays a potent antinociceptive effect in rodents and non-human primates without causing aversion, sedation, motor incoordination, or anhedonia in several pain models[25–27].

While the structural basis for KOR activation has been characterized by several studies providing high-resolution structures of both active-state and inactive-state KOR in complex with ligands[28–30], the structural mechanisms that underpin KOR-biased signaling are not fully understood. The promising pharmacology of nalfurafine and its G protein-biased activity make it a valuable tool to illuminate the

distinctive features responsible for biased signaling at KOR. Here, we use a combination of structure determination, molecular dynamics (MD) simulations, and functional assays to compare the molecular effects of nalfurafine on KOR to that of other KOR agonists with a range of pharmacological profiles. We determine the crystal structure of the human KOR in a complex with nalfurafine with an active state stabilizing nanobody (Nb39) at 3.3 Å resolution (Supplemental Table 1), which additionally provide insight into nalfurafine's subtype selectivity. We also identify a KOR selective arrestin-biased agonist, WMS-X600. By employing MD simulations, we identify molecular features related to preferential G protein activation or arrestin recruitment by observing the conformational changes in KOR upon stimulation by pharmacologically distinct agonists (G protein biased, balanced, arrestin biased). These findings shed light on the structural basis and dynamic features for KOR signaling bias, which provide an alternative rational for conformation-specific ligand design.

## Results
### Structure of nalfurafine-bound active-state of KOR

The nalfurafine's full agonist activity with high affinity and selectivity at KOR was first confirmed using radioligand binding and cell-based functional assays (Fig. 1a, b, Supplemental Table 2). The characterization also showed that nalfurafine has sub-μM affinity and is a partial agonist at other opioid receptors (Fig. 1a, b, Supplemental Table 2), although the therapeutic activity of nalfurafine is reportedly mediated by the KOR[31]. To better understand the atypical properties of nalfurafine, structural determination of the KOR-nalfurafine was first pursued to see if nalfurafine binds or signals differently from the other KOR agonists. The structure of KOR bound nalfurafine was determined at 3.3 Å using the previous KOR-MP1104-Nb39 construct with an additional thermostabilizing mutation identified by a computational approach[32] (Fig. 1c). This additional mutation, S324$^{7.47}$C, did not significantly change the expression level of the KOR, but increased the protein melting temperature (Tm) by 8 degrees compared to the wild

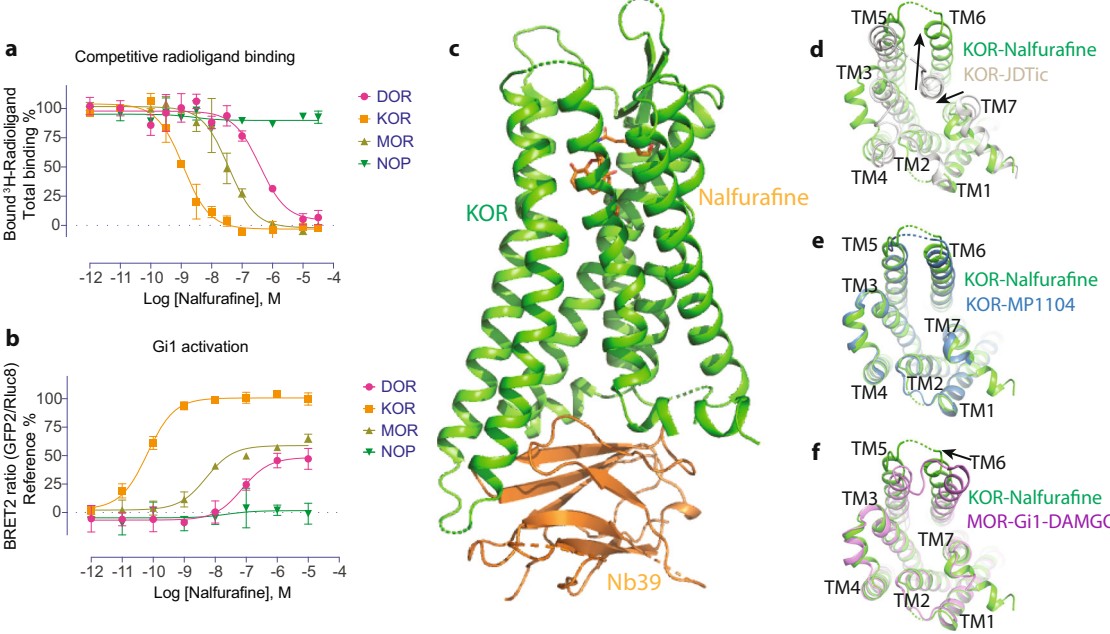

**Fig. 1 | Structure of active-state KOR bound to nalfurafine. a** The binding affinity of nalfurafine at four opioid receptors. $^3$H-Diprenorphine was used as radioligands for DOR, KOR, and MOR; $^3$H-Nociceptin was used as a radioligand for NOP. **b** The functional activity of nalfurafine at the opioid receptors. References were DPDPE, U50,488, DAMGO, and Nociceptin for DOR, KOR, MOR, and NOP, respectively. Data are expressed as the mean ± SEM of three independent experiments (*n* = 3

experiments each done in duplicate). **c** Structure of KOR-nalfurafine-Nb39 complex. **d** Comparison of active-state KOR-nalfurafine and inactive-state KOR-JDTic structures (intracellular view). **e** Comparison of active-state KOR-nalfurafine and KOR-MP1104 structure (intracellular view). **f** Comparison of KOR-nalfurafine and MOR-Gi1-DAMGO complex structures (intracellular view). Values in each plot were summarized in Supplemental Table 2.

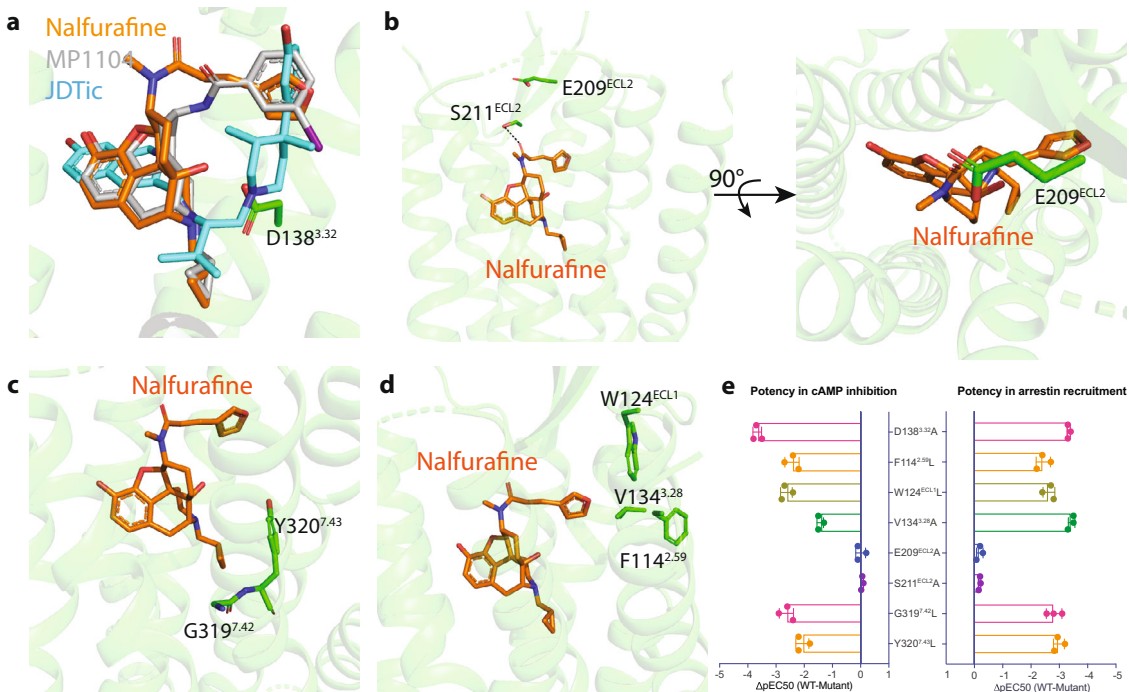

**Fig. 2 | Molecular signatures of nalfurafine-bound KOR. a** Alignment of the antagonist JDTic, agonist MP1104, and nalfurafine in the binding pocket of KOR. **b** The residue E209$^{ECL2}$ forms a lid on the top of nalfurafine. **c** The interaction between the cyclopropyl methyl group of nalfurafine and hydrophobic pocket residues. **d** The furan ring of nalfurafine forms unique interaction with the 'triad'

pocket including F114$^{2.59}$/W124$^{ECL1}$/V134$^{3.28}$. **e** Mutagenesis studies confirm the role of residues that interact with nalfurafine. Data are expressed as the mean ± SEM of three independent experiments ($n = 3$ experiments each done in duplicate) and summarized from curves attached in Supplemental Figs. 1–3. Values in each plot were summarized in Supplemental Tables 3–5.

type, indicating the S324$^{7.47}$C mutation increased the stability of the receptor (Supplemental Fig. 1a, b). Although not at the same position, the serine to cysteine mutation has been reported to improve the stability in both KOR[33] and δ-opioid receptor (DOR)[34]. It is worth mentioning that the functional characterization in the cAMP inhibition assay supported that nalfurafine activates both wild-type and S324$^{7.47}$C mutated KOR in a similar manner (Supplemental Fig. 1c, Supplemental Table 3). The global electron density, particularly the ligand binding pocket, enables us to build in the ligand with unambiguous orientation and conduct further analysis (Supplemental Fig. 1d).

The structure of KOR-nalfurafine displays an active-like state with characteristic outward movement of TM6 and inward movement of TM7 compared with the inactive KOR-JDTic structure (Fig. 1d). The overall structure of KOR-nalfurafine is similar to the previous active-state KOR-MP1104 structure (Fig. 1e), likely due to the application of the same nanobody (Nb39) in the complex assembly. However, the comparison between KOR-nalfurafine-Nb39 and MOR-DAMGO-Gi1[35] structures showed that TM6 adopts different orientations to accommodate the nanobody or transducer (Fig. 1f). MD simulations have also predicted such conformational differences between KOR-MP1104-Gi1 and KOR-MP1104-Nb39 structures[36]. The active-state stabilizing Nb39 or G proteins have been shown to behave as positive allosteric modulators for opioid receptors or other GPCRs[37]. Binding assays in the presence of either Nb39 or G proteins showed a significant increase in the binding affinity of nalfurafine but to different extents (40-fold with Nb39 and 10-fold with Gi1) (Supplemental Fig. 1e, Supplemental Table 3).

## Molecular recognition of nalfurafine by KOR

As a morphinan-scaffold ligand, nalfurafine shares parts of the binding pocket with previously determined MP1104-bound structures, such as an ionic-bridge interaction with the anchoring residue D138$^{3.32}$ (Supplemental Fig. 1f, g, Supplemental Table 3). Both binding poses are, however, clearly distinct from the pose of KOR antagonist JDTic

(Fig. 2a, Supplemental Fig. 2a–d). These differences provide a potential explanation for the antagonistic activity of JDTic and full agonistic activity of nalfurafine. Specifically, JDTic adopts a V-shaped conformation extending to the bottom of the binding cleft. This tight fit ensures that the ligand forms extensive ionic, polar, and hydrophobic interactions with the receptor (Supplemental Fig. 2b), resulting in the high binding affinity of JDTic and a long duration of action. Nalfurafine, instead, adopts a reversed V-shaped conformation and forms H-bond and hydrophobic interactions with the extracellular regions of KOR (Supplemental Fig. 2d). A similar reversed V-shape is also observed in the MP1104-KOR binding pose (Supplemental Fig. 2c). These interactions induce unique conformational changes of the receptor and together contribute to the agonist activity of nalfurafine.

First, the extracellular loop (ECL) 2 moves ~8 Å to adopt the inactive state and forms an apparent lid on top of nalfurafine (Fig. 2b, Supplemental Fig. 2a). This is a feature that has been frequently observed in many other GPCR structures, particularly those with slow off-rate ligands like lysergic acid diethylamide (LSD)-bound 5-HT2B serotonin receptor[38] and risperidone-bound D2 dopamine receptor[39]. This movement appears to be ligand-dependent because the ECL2 adopts a different orientation between active and inactive states (Supplemental Fig. 2a). Another ECL2 residue S211$^{ECL2}$ forms strong H-bond interaction with the carbonyl oxygen that contributes to the residence time of nalfurafine as the mutants (S211$^{ECL2}$A and the lid residue E209$^{ECL2}$A) accelerate the dissociation rate of nalfurafine without affecting its agonist activity (Fig. 2e, Supplemental Fig. 2e, f, Supplemental Table 4).

Second, the cyclopropyl methyl group of nalfurafine, as well as in MP1104, extends to a deeper pocket compared to the isopropyl group in JDTic (Fig. 2a, c). Our previous studies on MP1104 have shown that interactions with this hydrophobic pocket are important for the agonist activity at KOR[29]. Mutational studies also support that the residues in the hydrophobic pocket significantly decrease the potency of nalfurafine (Supplemental Fig. 2g, Supplemental Table 4). These

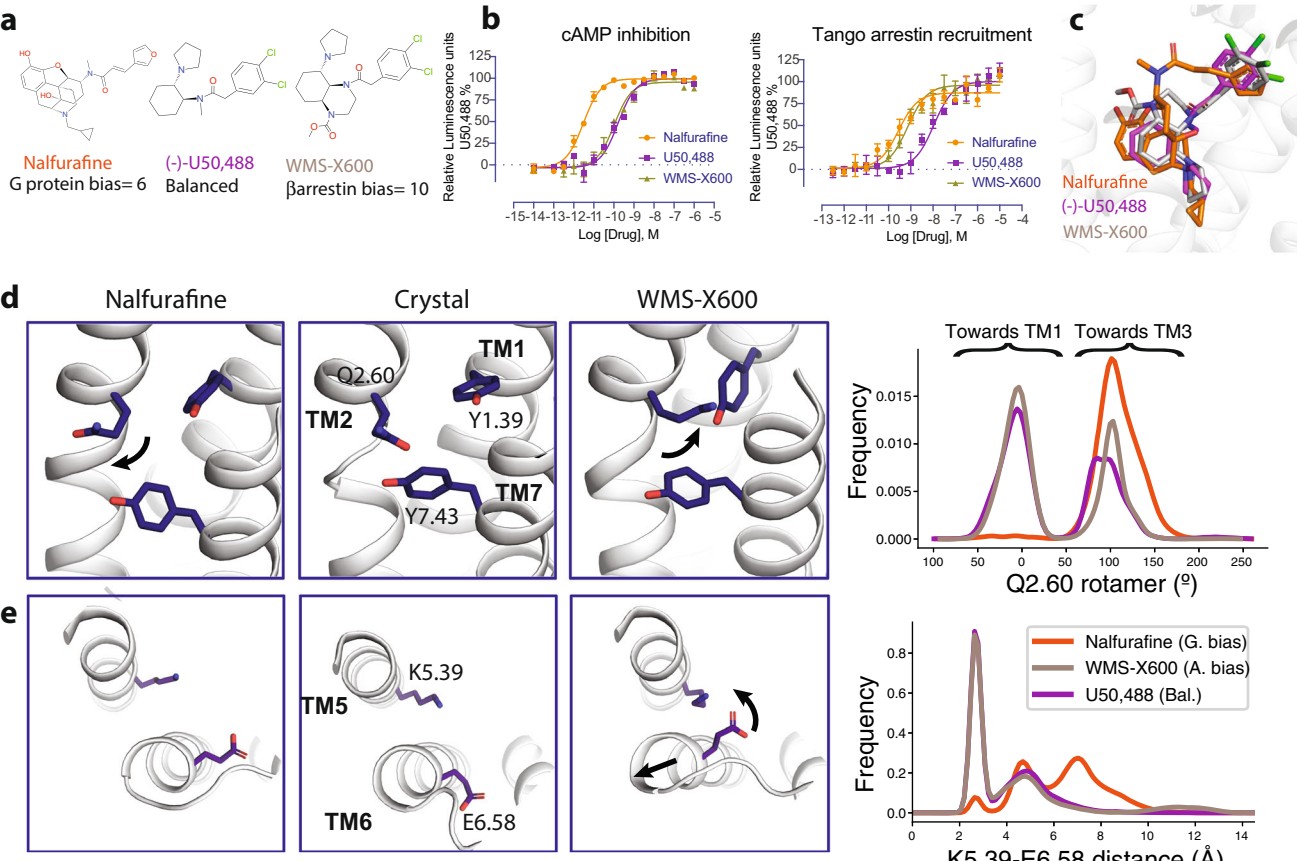

**Fig. 3 | Molecular dynamics simulations of functionally selective KOR agonists.**
**a** Chemical structure of G protein biased agonist nalfurafine, balanced agonist U50,488, and arrestin-biased agonist WMS-X600. U50,488 was used as a reference agonist that has a bias factor = 1; G protein bias factor of nalfurafine (95% confidence interval) = 6 (4.5–8.2); arrestin2 bias factor of WMS-X600 (95% confidence interval) = 10 (6.5–15.4). Calculation of bias factor was described in the "Methods".
**b** Functional characterization of KOR agonists in G protein-mediated cAMP inhibition and arrestin-mediated recruitment. Data are expressed as the mean ± SEM of three independent experiments ($n = 3$ experiments each done in duplicate).
**c** Overlay of nalfurafine, U50,488, and WMS-X600 in the binding pocket of KOR.
**d** Differences in Q115[2.60] rotamer orientations favored by different ligands. In the left panel, Q115[2.60] points towards TM3, keeping the pocket close to its starting configuration (nalfurafine). In the right panel, the rotation of Q115[2.60] out of the pocket depresses Y66[1.39], allowing Y320[7.43] to move forward and the top of TM7 to rotate counter-clockwise (WMS-X600). The Q115[2.60] rotamer is quantified as the dihedral angle formed by the C, Ca, Cg, and Cd atoms of Q115[2.60]. **e** Differences in TM5 and TM6 conformation favored by different ligands. The distribution of K227[5.39] amine nitrogen to E297[6.58] carboxylate oxygen distances shows that the two residues generally do form a salt bridge when WMS-X600 or U50,488 is bound but do not when nalfurafine is bound. Values in each plot were summarized in Supplemental Table 6.

hydrophobic residues appear to play a more important role in nalfurafine-mediated arrestin recruitment as both the potency and efficacy are significantly decreased or abolished (Supplemental Fig. 2h, Supplemental Table 4). The agonist-mediated signal transduction is likely achieved by the bulky cyclopropylmethyl group that sterically pushes a tryptophan W287[6.48] away and results in a more downward movement of the side chain of W287[6.48] (4.0 Å, Supplemental Fig. 3a); the latter is often referred to as the "rotamer toggle switch." Mutation of the W287[6.48] to alanine significantly reduces the arrestin recruitment of nalfurafine, but with minimal effect on the G protein-mediated activation (Supplemental Fig. 3b, Supplemental Table 5). This may also support the antagonist mechanism of JDTic because the dimethyl group maintains a strong hydrophobic interaction with the W287[6.48] (3.5 Å, Supplemental Fig. 3a). Studies on different MOR agonists (e.g., biased, balanced, partial, and antagonist) also identified ligand-specific binding poses with respect to W[6.48], suggesting a key role of this residue in mediating ligand signaling efficacy[10].

Third, the major difference between nalfurafine and other ligands comes from its unique furan ring. The furan ring points to the crevice between TM1 and TM2, leading to the conformational changes by pushing the hydrophobic residue V134[3.28] and the bulky aromatic residues like W124[ECL1] and F114[2.59] (Fig. 2d, Supplemental Fig. 3c). This

movement appears to be important for receptor activation as mutation of residues in this triad significantly decreases the potency of nalfurafine and almost abolishes the arrestin recruitment activity (Fig. 2e, Supplemental Fig. 3d, Supplemental Table 5).

### Nalfurafine is G protein-biased and WMS-X600 is arrestin-biased relative to U50,488

Nalfurafine's therapeutic property compared to other KOR agonists inspired us to identify the structural and molecular support for its unique pharmacology. While U50,488 (Fig. 3a) has been known as a KOR balanced agonist, nalfurafine has been reported by several studies to display moderate to significant G protein bias activity[26,40]. We confirmed that nalfurafine preferentially activates G protein pathway using a luminescence-based reporter assay (bias factor = 6, bias factor (U50,488) as a reference = 1, see "Methods" for bias factor calculation) (Fig. 3b, Supplemental Table 6). To better understand the G protein vs. arrestin signaling mechanism, we also identified the first known arrestin-biased KOR agonist, WMS-X600. In particular, we found that WMS-X600, which was previously designed based on the U50,488 scaffold through structure-activity relationship (SAR) analysis[41], has a bias factor of 10 toward the arrestin pathway relative to U50,488 (Fig. 3b). The functional selectivity of these KOR agonists was

further validated by a secondary Bioluminescence Resonance Energy Transfer (BRET) assay (Supplemental Fig. 4a, Supplemental Table 7). With the three different types of ligands, it enables us to conduct a comprehensive analysis into the molecular basis of biased signaling at KOR.

## Receptor–ligand interactions that contribute to biased signaling at KOR

We used molecular dynamics simulations to investigate differences in the receptor conformational ensemble favored by nalfurafine, U50,488, and WMS-X600. Since these ligands have a range of signaling profiles, observed conformational differences were used to form hypotheses about what conformations result in biased signaling, which we then validated with mutagenesis experiments.

Our simulations were initiated from the KOR-nalfurafine-Nb39 structure described here. Prior to initiating simulations, we removed the nanobody so that the intracellular side of the receptor is free to adopt different conformations. We used molecular docking to generate poses for U50,488 and WMS-X600, since there are no experimentally determined structures of KOR bound to these ligands (see "Methods"). This docking analysis revealed that all three ligands can adopt similar binding poses: forming a salt bridge with D138[3.32], placing a hydrophobic group in the central pocket, and positioning an aromatic ring between TM2 and TM3 (Fig. 3c). However, MD simulations revealed that subtle differences in receptor–ligand interactions result in the ligands favoring different conformational ensembles of the receptor (Supplemental Fig. 4b). In this section, we report the direct effects on the binding pocket. In the following sections, we describe how these effects propagate to the intracellular transducer coupling interface.

First, in simulations with nalfurafine bound, the side chain of residue Q115[2.60] tends to stay close to the crystallographic conformation, whereas with U50,488 and WMS-X600 bound, Q115[2.60] frequently rotates away from the ligand, towards TM1 (percentage of simulation time in which Q115[2.60] is rotated towards TM1: 2%, 58%, 61%, respectively; the difference between nalfurafine and each of the other ligands is significant with $p = 0.002$ (Fig. 3d). Two factors appear to cause this difference: the connection between nalfurafine's pendant ring extends farther towards the extracellular region and TM5, and the furan moiety on nalfurafine's pendant ring can form hydrogen bonds with the side chain of Q115[2.60], whereas the pendant rings of WMS-X600 and U50,488 lack hydrogen-bonding partners. Supporting our hypothesis that Q115[2.60] forms favorable interactions with nalfurafine and unfavorable clashes with WMS-X600, in vitro experiments show that a Q115[2.60]N mutation reduces the potency of nalfurafine by 5-fold but increases the potency of WMS-X600 for G protein and arrestin signaling by 10-fold and 5-fold, respectively (Supplemental Fig. 4c, Supplemental Table 7).

Second, nalfurafine, but not U50,488 or WMS-X600, destabilizes the ionic bond between K227[5.39] and E297[6.58] (Fig. 3e). We hypothesize that this bond is rarely formed in nalfurafine-bound KOR simulations because the methyl group on the amine linking nalfurafine's morphinan scaffold with its pendant ring sterically hinders K227[5.39] from interacting with E297[6.58] (Fig. 3e). Supporting the importance of the K227[5.39]–E297[6.58] salt bridge for arrestin recruitment, functional assays of nalfurafine with K227[5.39] mutated to alanine, show reduced arrestin coupling, despite increased G protein coupling (Supplemental Fig. 4d, Supplemental Table 7), and an increased distance between the extracellular ends of TM5 and TM6 for nalfurafine in MD simulations (Supplemental Fig. 4e).

Third, WMS-X600 induces a downward (towards intracellular) translation of W287[6.48] as compared to U50,488 (87 and 46% of simulation time, respectively, $p = 0.02$) (Supplemental Fig. 5a–c). Compared to U50,488, WMS-X600 contains an additional ring, which provides a bulkier and more rigid group in the central pocket,

impacting the position of W287[6.48]. This is the only difference between the binding pocket dynamics with U50,488 or WMS-X600 bound that we were able to identify. To test whether this difference could explain the arrestin-bias of WMS-X600, we characterized the effects of a W287[6.48]A mutation. We hypothesized that if the key contributor to WMS-X600's arrestin bias is that it presses downward on W287[6.48], then making this residue smaller should reduce its arrestin bias. Indeed, for WMS-X600, we observed that the W287[6.48]A mutation leads to an 82-fold loss of potency in G protein activation (1.32 nM in WT vs. 108 nM in W287[6.48]A) and a 232-fold loss of potency in arrestin recruitment (1.28 nM in WT vs. 298 nM in W287[6.48]A) (Supplemental Fig. 5d, Supplemental Table 8). Meanwhile, for U50,488, we observed a 54-fold decrease of potency in G protein activation (3.91 nM in WT vs. 210 nM in W287[6.48]A) and only an 8-fold decrease in arrestin recruitment (43.8 nM in WT vs. 375 nM in W287[6.48]A) (Supplemental Fig. 5e, Supplemental Table 8). In addition, nalfurafine is similar to WMS-X600 in that it often induces a downward translation of W287[6.48] and the W287[6.48]A mutation has a much larger effect on arrestin recruitment than on G protein activation (Supplemental Fig. 3b, Supplemental Table 5).

## Nalfurafine favors a unique conformation of the transducer-coupling interface

In our simulations, the intracellular part of the receptor adopts three major conformational states (Fig. 4a): (1) the canonical active state, i.e., the conformation most commonly observed in experimental structures of GPCRs in complex with either G proteins or arrestins, (2) the alternative state, which we previously observed to be associated with arrestin-biased signaling at Angiotensin II type 1 receptor (AT1R)[12], and (3) what we will refer to as the "occluded state". The occluded state is characterized by the very bottom of TM7 rotating clockwise and moving towards TM2. At KOR, the alternative state is similar to that observed at AT1R, except that we do not observe a downward rotamer of Y330[7.53].

These three states are occupied in different proportions when each of the ligands is bound. In simulations with the balanced agonist U50,488 bound, the receptor generally adopts the canonical active conformation (Fig. 4b, Supplemental Fig. 6a). In simulations with the arrestin-biased ligand WMS-X600 bound, the receptor spends a significant portion of time in the alternative state (37% as compared to 2% for U50,488; $p = 0.03$). This is consistent with our previous work showing that the alternative state is associated with arrestin-biased signaling at AT1R[12] and MOR[42]. However, the functional selectivity of nalfurafine can not be explained in a similar manner, as nalfurafine induces the alternative state for a comparable amount of time as WMS-X600 (41% vs. 37%, respectively). However, most of the remaining time is spent in the occluded state (36% as compared to 4% for U50,488 and 6% for WMS-X600; $p = 0.02$ and $p = 0.03$, respectively). This leads us to hypothesize that occupation of the occluded state could be responsible for nalfurafine's G protein bias.

## Mechanism of allostery between the binding pocket and the intracellular interface

How do the direct effects of the ligands on the binding pocket propagate to the intracellular transducer coupling interface? The differences in ligand interactions with Q115[2.60] and W287[6.48] act together to control the rotation of TM7. The rotamer of Q115[2.60] is coupled to the rotamer of Y66[1.39], which itself is coupled to a rotation of TM7 in the vicinity of Y320[7.43] by Y66[1.39]'s stabilizing hydrogen bond with the backbone oxygen of Y320[7.43] (Fig. 3d). The influence of W287[6.48] is mediated by a direct interaction between the N322[7.45] side chain and the W287[6.48] indole nitrogen (Supplemental Fig. 5c).

The above factors largely determine the rotation of TM7 in the connector region and thereby control the relative occupancy of the alternative and canonical states. However, these factors can not

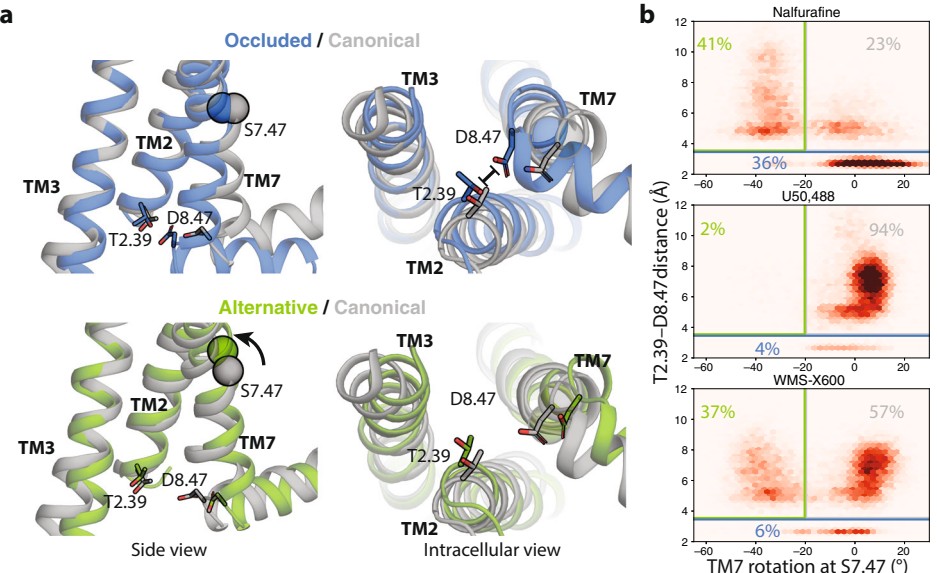

**Fig. 4 | Nalfurafine, U50,488, and WMS-X600 favor different receptor conformations. a** Side and intracellular views of representations of the occluded (blue) and alternative state (green) compared to the canonical state (gray). Here the alternative and occluded states are represented by simulation frames and the canonical state is represented by the KOR-nalfurafine-Nb39 crystal structure.

**b** Density plots of the distance between the carboxylate oxygens of D334[8.47] and the hydroxyl oxygen of T94[2.39] and the rotation of TM7 at the S324[7.47] (see "Methods"). Simulation frames were classified as being in the occluded state if the T94[2.39]–D334[8.47] distance is less than 3.5 Å and otherwise in the alternative state if the TM7 rotation is less than −20° or else in the canonical active state.

completely explain the occupancy of the occluded state. With nalfurafine bound, a clockwise or neutral rotation in the connector region generally results in the receptor adopting the occluded state, whereas when other ligands are bound, the receptor generally remains in the canonical state. We speculate that this is due to the unique effect of nalfurafine on the K227[5.39]–E297[6.58] salt bridge, as this is the most prominent difference in binding pocket dynamics apart from the rotamer of Q115[2.60]. Disruption of the K227[5.39]–E297[6.58] salt bridge by nalfurafine results in an increased distance between the extracellular ends of TM5 and TM6 as compared to what is observed for U50,488 or WMS-X600 (Supplemental Fig. 4e). It is likely that this difference propagates to the intracellular part of the receptor and contributes to ligand-specific transducer coupling, as the 6′-Guanidinonaltrindole (6′-GNTI), another G protein-biased KOR agonist[43], has a positively charged guanidinium group that likely forms an ionic bond with E297[6.58] and disrupts the salt bridge as nalfurafine (Supplemental Fig. 6b).

**Ligand-specific transducerome profiling**

KOR signals through seven different G protein subtypes (Gi1, Gi2, Gi3, GoA, GoB, Gz, and Gustducin) and two β-arrestins (β-arrestin 1 and β-arrestin 2). The different conformations induced by the ligands, particularly the intracellular end as observed in the MD simulations, suggest that they may have different transducer coupling preferences. A detailed comparison of the signaling profile using the TRUPATH assay[44] was performed to systematically examine these intracellular transducer-coupling preferences (Fig. 5). U50,488 (a balanced KOR agonist) was used as the reference ligand, and efficacy for nalfurafine and WMS-X600 were expressed as a percentage of the U50,488 response at each transducer.

The data were presented in the order of potency from high to low. It is found that all three KOR agonists have the highest potency at the pertussis toxin-insensitive Gz, instead of Gi or Go subtypes (Fig. 5a, Supplemental Table 9). A similar phenomenon has also been observed in MOR and other Gi/o coupled receptors[45], which suggests a ligand-specific transducer coupling. For the other G protein subtypes, each ligand similarly activates the Gi/o sub-family (Gi1, Gi2, Gi3, GoA, and

GoB), which is consistent with their high sequence identity. Gustducin is the endogenous transducer for the taste receptors, such as TAS2R. Our data showed that KOR robustly activates Gustducin but displays ligand-specific patterns. While U50,488 and WMS-X600 both potently activate Gustducin (EC50 = 59.9 nM and 4.9 nM, Efficacy = 99% and 94%, respectively), nalfurafine partially activates it (EC50 = 3.5 nM, Efficacy = 65%). This is also observed in the test of 6′-GNTI[43], with no detectable activation of Gustducin. Coincidentally, the endogenous ligand of KOR, dynorphin A 1–13, was shown as an efficacious agonist at most transducers but nearly inactive at Gustducin[44]. In terms of arrestin recruitment, WMS-X600 displays full agonism at both β-arrestin 1 and β-arrestin 2 recruitment as the efficacy is similar to U50,488 (Fig. 5a, Supplemental Table 9). In potency, both WMS-X600 and U50,488 have a slight preference for β-arrestin 2. Nalfurafine displays partial agonist activity at both arrestin subtypes: 80% at β-arrestin 2 and 50% at β-arrestin 1. Consistent with this observation, the KOR G protein-biased agonist 6′-GNTI is also a partial agonist at β-arrestin 2 recruitment (10%), and has no detectable efficacy at β-arrestin 1 (Fig. 5a, Supplemental Table 9). Together with a similar interaction and state induced between nalfurafine and 6′-GNTI, it suggests a convergent mechanism for G protein-bias at KOR.

Similar to what we observed in molecular simulations that particular residues differentially affect G protein and arrestin signaling, mutagenesis screening also identified molecular determinants including residues or motifs responsible for G protein subtype activity (Fig. 5b, Supplemental Figs. 7, 8, Supplemental Tables 10, 11). These tested mutations did not significantly (<2-fold) alter the receptor expression as examined by radioligand binding assays (Supplemental Fig. 8). R170[ICL2] is a residue that is conserved in opioid receptors and has been shown to participate in the interaction with the helix α5 of Gi1[35]. Mutation of R170[ICL2] to alanine in KOR decreases the potency of nalfurafine about 100-fold at Gi1, Gi2, and Gi3, and 10-fold at other G protein subtypes (Supplemental Fig. 7, Supplemental Tables 10, 11). D155[3.49] is involved in a highly conserved motif (D[3.49]R[3.50]Y[3.51]) in class A GPCRs and conformational changes of this motif have been reported as an indicator of GPCRs activation. The mutation D155[3.49]A decreases nalfurafine's potency at Gi1 nearly 1000-fold compared to other G

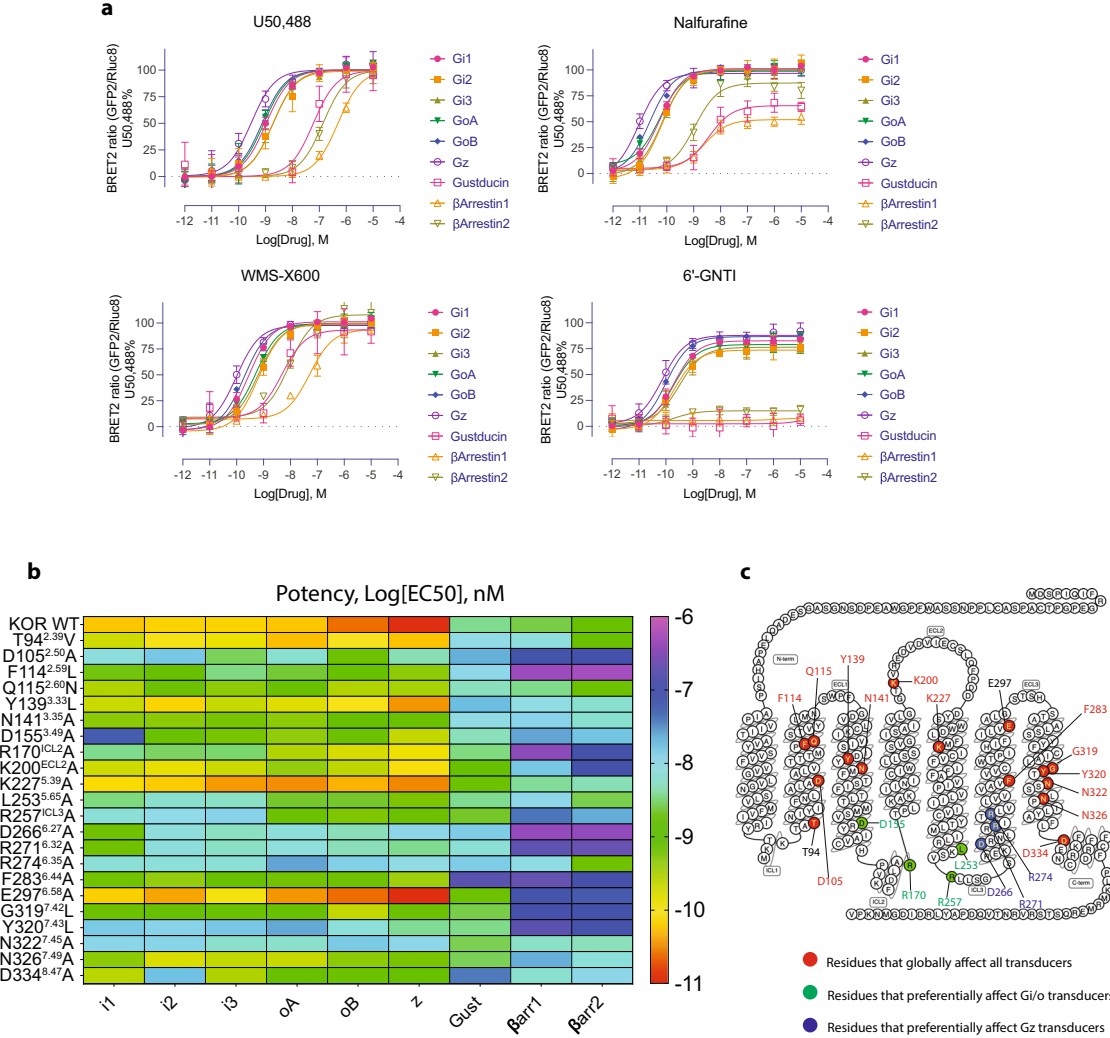

**Fig. 5 | The functional characterization of ligand-specific transducer coupling. a** The functionally selective KOR agonists display preference towards unique transducer coupling. Data are expressed as the mean ± SEM of three independent experiments ($n = 3$ experiments each done in duplicate). **b** The mutagenesis screening represented by the heatmap confirms that nalfurafine-specific transducer coupling is affected by specific residues of the receptor. **c** Positions of residues in nalfurafine-mediated transducer coupling. Values in each plot were summarized in Supplemental Tables 9–11.

protein subtypes (<10-fold) (Supplemental Fig. 7, Supplemental Tables 10, 11). Both R170[ICL2] and D155[3.49] have also been shown to play a crucial role in coupling to the Gi1 by forming direct charge-charge interaction with the Gi1 residue D350 using MD simulation[36]. L253[5.65]A and R257[ICL3]A both selectively decrease the Gi subtypes (Gi1, Gi2, and Gi3) by 100-fold with 10-fold on other G protein subtypes (Supplemental Fig. 7, Supplemental Tables 10, 11). Several other residues globally affect the G protein coupling, but a detailed comparison showed they have more effects on Gz coupling than the other subtypes.

Regarding the mutational effects on arrestin recruitment, we observed that most of the mutations caused a significant loss of potency in nalfurafine-mediated β-arrestin 1 and 2 recruitment, suggesting that KOR-arrestin interface involves more residues than the KOR-G protein interface, as identified in 5-HT2B-Gq and 5-HT2B-β-arrestin1 structures[46]. We also observed that most of these residues are located in the intracellular regions (ICLs and helix end) where the receptor and G protein form most interactions. These residue-specific effects on G protein coupling also support that the coupling of individual G protein subtypes is likely determined by both conformation of the receptor and their unique interaction with the intracellular-end residues (Fig. 5c).

Recent structures of the MOR-Gi[35], neurotensin-arrestin[47], and M2-arrestin complexes[48] provide an explanation for how the occluded state could result in specific transducer coupling. In models of KOR-arrestin complexes, an aspartate on the finger loop binds to T94[2.39] (Supplemental Fig. 9). In the occluded state, D334[8.47] binds to this same position, perhaps competitively disfavoring arrestin coupling. Unlike the finger loop, the α5 helix of Gi avoids these charge conflicts. The carboxy-terminus of the helix is pointed towards TM6 in the structure of the MOR–Gi1-protein complex, and the only other negative residue, D350, is pointed downwards towards the solvent and away from the TM7/TM2 interface. In addition, the α5 helix has a glycine in the spot closest to TM7, minimizing steric clashes between TM7 and the G-protein as TM7 moves towards TM2. Taking these modeling data together, the occluded state may disfavor arrestin coupling while still permitting G-protein coupling.

## Discussion

Activation of KOR mediates both non-addictive analgesia and hallucinogenic or dysphoric activities. Correlations between unique signaling pathways and pharmacological outcomes are still under debate due to an incomplete structural and mechanistic understanding of KOR agonist actions. Here, we presented the crystal structure of KOR bound

to its first clinically available drug, nalfurafine. With the combination of structural biology, molecular dynamics simulation, and mutagenesis validation, we provide molecular evidence into ligand-specific conformations of KOR that correlate with the preferential signaling mediated by G protein-biased, balanced, and arrestin-biased agonists, respectively. Considering the unique role of KOR in pain management —agonists that activate KOR do not produce addictive side effects—the specific conformation stabilized by nalfurafine may provide guidance for designing safer opioid analgesics.

We identified specific interactions with the furan ring of nalfurafine that are essential for its agonist activity. The binding of nalfurafine induces conformational changes of the ECL2, leading to the formation of a strong hydrogen bond with S211[ECL2]. Interactions with the extracellular helices and loops have been shown to restrict the conformational changes and thus affect G protein or arrestin signaling[38,49–51]. This interaction appears to further stabilize the binding of nalfurafine in the pocket as mutation of this serine to alanine accelerates the dissociation of nalfurafine from the pocket. Due to the long amide arm, the furan ring of nalfurafine extends further away than the iodinated phenyl ring of MP1104. We discovered that a hydrophobic triad F114[2.59]-W124[ECL1]-V134[3.28] was pushed away by the furan ring and this movement is important for nalfurafine's agonist activity as confirmed by the mutagenesis studies.

Previous studies have found that nalfurafine displays moderate to significant G protein bias upon KOR activation compared to the balanced agonist U50,488. Using two independent assays (cAMP inhibition vs. Tango arrestin; BRET-G protein dissociation vs. BRET-arrestin recruitment), we found that nalfurafine exhibits a preference toward the G protein signaling over arrestin pathway. In addition, nalfurafine is a partial agonist in KOR-mediated arrestin recruitment confirmed by the BRET assays. Gillis et al. have previously suggested that the reduced side effects profiles for previously reported G-protein biased MOR agonists, such as TRV130, PZM21, and SR-17018 that possess low intrinsic efficacies compared with drugs such as fentanyl may be correlated with partial agonism rather than biased signaling per se[52].

Although the structure of KOR bound to nalfurafine is of similar conformation to the MP1104-bound KOR, MD simulations provide evidence that functionally distinct ligands induce specific conformational changes in the receptor. We found that the arrestin-bias of WMS-X600 relative to U50,488 can be explained by different occupancies of the previously identified canonical and alternative states, consistent with our past work at AT1R[12] and MOR[42]. In this study, we identified the "occluded state", which shows an inward movement of TM7 in the intracellular coupling interface, which is observed far more often in simulations with nalfurafine bound than with balanced U50,488 or arrestin-biased WMS-X600 bound.

The observed ligand-specific differences in binding pocket dynamics are consistent with prior work and explains the functional selectivity of other KOR ligands. We previously showed that ligands placing bulky groups in the TM2–TM3 sub-pocket favor arrestin signaling, whereas leaving this sub-pocket unoccupied favors G protein signaling[6]. Leaving the TM2–TM3 sub-pocket unoccupied would likely have a similar effect on Q115[2.60] as what we observe in nalfurafine-bound simulations, suggesting that nalfurafine and our previously reported ligands achieve G protein bias through a common mechanism. Moreover, we previously observed a similar relationship between a ligand's bias profile and its effect on the orientation of Q2.60 at MOR[42]. At MOR, the G protein-biased ligand mitragynine pseudoindoxyl positions a ring between TM7 and Q2.60, sterically constraining it in a similar orientation as we most commonly observe with nalfurafine bound, whereas the arrestin-biased ligand lofentanil generally favors Q115[2.60] being oriented towards TM7, as we observe with U50,488 and WMS-X600. Similar studies using NMR[10] and MD simulations[53] showed the MOR G-protein biased agonists also induce specific conformations

in the TM7, ICL1, and H8 areas, and this conformation is maintained after binding to a G protein.

We further provide experimental validation that specific residues or motifs differentially affect intracellular transducer coupling, not only G protein vs. arrestin but also Gi/o vs. Gz vs. Gustducin. Gz has been found to be predominantly present in neurons in the brain. Although the activation of Gz inhibits cAMP production similarly to other Gi/o proteins, several findings have suggested its unique role in cellular communication[54,55]. It is worth pointing out that we used the TRUPATH assays in KOR-expressing cell lines to comprehensively examine these intracellular transducer-coupling preferences. Although G protein signaling-biased agonism at KOR is maintained in striatal neurons[56], our studies could be more informative by extending to the physiological conditions using neuronal cell line or freshly isolated neurons.

Biased ligands of KOR offer promising strategies to enhance desirable analgesia while reducing side effects. Here, our KOR-nalfurafine structure serves as the starting point for molecular dynamics simulations that shed light on which protein interactions and motions may account for KOR's therapeutic potential. We propose conformations in the binding pocket and TM7 that differentiate nalfurafine from other examined balanced or arrestin-biased ligands as well as provide a basis for studying biased signaling at KOR in general. Continued insight into biased signaling at KOR is a favorable avenue for developing safer analgesics and lessening the blight of the opioid epidemic.

## Methods

### Human KOR receptor crystallization construct generation

An engineered receptor construct generated by Quickchange PCR was employed to perform the crystallization of the human KOR complex. The final construct lacks both the N-terminal residues (1–53) and the C-terminal residues (359–380). In addition, the receptor N-terminus residues M1-H53 was replaced by A1-L106 of the thermostabilized apocytochrome b$_{562}$ RIL (BRIL) from *E. coli* (M7W, H102I, R106L). To facilitate crystallization, a glycine-serine linker was inserted between BRIL and receptor. Additional modifications were considered to enhance thermostability by introducing two mutations at I135L and S324C as well as facilitate purification by immobilized metal affinity chromatography by having a haemagglutinin (HA) signal sequence, then a FLAG tag at the N terminus, followed by a 10× His tag, then a TEV protease site.

### KOR expression and purification

To obtain high-titer recombinant baculovirus (>10$^9$ viral particles per ml), Bac-to-Bac Baculovirus Expression System (Invitrogen) was utilized. In brief, recombinant bacmid (~5 μg) and 3 μl of a lipid suspension, Cellfectin II Reagent (Invitrogen), were incubated for 30 min into two separate 50 μl of Sf-900 II SFM media (Invitrogen). The above mixtures were transfected into 400 μl Sf-900 II SFM media that included 5 × 10$^5$ settled *Spodoptera frugiperda* (Sf9) insect cells (Expression Systems) in a 12-well plate (Corning) to yield recombinant baculovirus. After 5 h, the media were replaced by 1 ml Sf-900 II SFM media (Invitrogen) and incubated at 27 °C for 5 days. After harvesting, P0 viral stock with ~10$^9$ virus particles per ml was obtained as the supernatant. High-titer baculovirus stock was then generated by the infection of 40 ml of Sf9 cells (cell density: 2–3 × 10$^6$ cells/ml) and its incubation for 3 days. A flow-cytometric analysis of cells stained with gp64-PE antibody was used to determine viral titers (Expression Systems). For KOR expression, Sf9 insect cell cultures were grown to a density of 2.5 million cells/ml and was then infected in ESF921 media (Expression Systems) with P1 or P2 virus at a MOI (multiplicity of infection) of 3.5% production boost additive (PBA, Expression Systems) was added to maintain cell alive. To support the receptor trafficking, naltrexone (10 μM, final concentration) was also added. After

48 h, centrifugation was applied to harvest the infected cells followed by washing with 1x PBS, and then stored at −80 °C for future use. The frozen cell pellets were resuspended in a low-salt buffer (10 mM HEPES, pH 7.5, 10 mM $MgCl_2$, 20 mM KCl containing protease inhibitors (500 μM AEBSF, 1 μM E-64, 1 μM Leupeptin, 150 nM Aprotinin)). Centrifugation was repeated 4 times in a high osmolarity buffer (1.0 M NaCl, 10 mM HEPES, pH 7.5, 10 mM $MgCl_2$, 20 mM KCl) for the purification of membranes. Subsequently, purified membranes were flash-frozen in liquid nitrogen and stored at −80 °C until use.

Upon use, purified membranes were resuspended in a buffer (10 mM HEPES, pH 7.5, 10 mM $MgCl_2$, 20 mM KCl, 150 mM NaCl, 50 μM nalfurafine, and 1x protease inhibitors (500 μM AEBSF, 1 μM E-64, 1 μM Leupeptin, 150 nM Aprotinin)) and then incubated at room temperature for 1 h. After that, the sample was kept at 4 °C for 30 min, and then 2 mg/ml iodoacetamide (Sigma) were added and incubated for 30 min further. Membranes were solubilized in 10 mM HEPES, pH 7.5, 150 mM NaCl, 1% (w/v) n-dodecyl-β-D-maltopyranoside (DDM, Anatrace), 0.2% (w/v) cholesteryl hemisuccinate (CHS, Sigma), and in protease inhibitors for 2 h at 4 °C. Centrifugation at $150,000 \times g$ for 30 min was applied to obtain the supernatant, which was then incubated with 20 mM imidazole and (500 μl per 1 L culture volume) TALON IMAC resin (Clontech) overnight at 4 °C for protein purified from 1 L of cells. The next day, 10 column volumes (cv) of wash Buffer I (50 mM HEPES, pH 7.5, 800 mM NaCl, 0.1% (w/v) DDM, 0.02% (w/v) CHS, 20 mM imidazole, 10% (v/v) glycerol, and 25 μM nalfurafine) was used to wash the resin, followed by 10 cv of Wash Buffer II (25 mM HEPES, pH 7.5, 150 mM NaCl, 0.05% (w/v) DDM, 0.01% (w/v) CHS, 10% (v/v) glycerol, and 25 μM nalfurafine). Proteins were eluted and concentrated to 500 μl using Wash Buffer II (2.5 cv) supplemented with 250 mM imidazole and 100 kDa molecular weight cut-off Vivaspin 20 concentrator (Sartorius), respectively. His-tagged TEV protease (Homemade) was added and incubated overnight at 4 °C to remove the N-terminal 10× His-tag. The suspension was passed through equilibrated TALON IMAC resin (Clontech) and the flow-through was collected to remove protease, cleaved His-tag, and uncleaved protein. The protein sample was incubated with excessive Nb39 (KOR/Nb39 m/m: 1:2) for 3 h. κOR-nalfurafine-Nb39 complexes were then applied to a 100 kDa molecular weight cut-off Vivaspin 500 centrifuge concentrator (Sartorius Stedim) to concentrate it to ~30 mg/ml. By using size-exclusion chromatography, protein purity, and monodispersity were analyzed.

### Lipidic cubic phase crystallization
By employing the twin-syringe approach[57], KOR-nalfurafine-Nb39 complexes were reconstituted into lipidic cubic phase (LCP) by mixing protein solution and a monoolein/cholesterol (10:1 w/w) mixture in a 2:3 protein solution to lipid ratio of (v/v). For crystallization, aliquots of the protein-LCP mixture were dispensed onto the 96-well LCP glass sandwich plate (Marienfeld GmbH) and overlaid with the precipitant solution using (Art Robbins Instruments). Following optimization, KOR-nalfurafine-Nb39 crystals were obtained in 100 mM Bis-tris pH 6.5−7.0, 140−200 mM magnesium sulfate hydrate, 100 mM sodium citrate tribasic dehydrate, and 10 mM Manganese (II) chloride tetrahydrate, 28−30% PEG400. After three days, the crystals were grown to full size (50 μm × 30 μm × 20 μm). MiTeGen micromounts were used to directly harvest the crystals from the LCP matrix, followed by flash-cooling and then stored in liquid nitrogen.

### Data collection, structure solution, and refinement
By using a 10 μm minibeam at a wavelength of 1.0330 Å and an Eiger detector, X-ray data were collected at the 23ID-B and 23ID-D beamline (GM/CA CAT) at the Advanced Photon Source, Argonne, IL. An unattenuated beam using 0.2° oscillation per frame for 0.2 s was applied to the crystals to collect diffraction data. The data obtained from 21 crystals were subjected to indexing, integrating, scaling, and merging using HKL3000[58]. Three independent models of a truncated 7TM

portion of the receptor and a nanobody Nb39 from the μOR-Nb39-BU-72 complex (PDB ID: 5C1M), and the thermostabilized apocytochrome $b_{562}RIL$ protein (PDB ID: 6B73) were processed with a molecular replacement in PHASER crystallographic software[59] to determine initial phases. In asymmetric unit, two copies of the 7TM portion of each receptor and the nanobody Nb39 without BRIL were identified. PHENIX[60] followed by manual examination and rebuilding of the refined coordinates in the program COOT[61] using $2mF_o$ - $DF_c$ and $mF_o$ - $DF_c$ maps were used for refinement. The data collection and structural refinement statistics are summarized in Supplemental Table 1.

### cAMP inhibition assays
To do G protein-mediated cAMP inhibition, human KOR in pcDNA3.1 and another plasmid expressing Glosensor luciferase (Promega) were used. HEK 293T cells (ATCC, CRL-3216) were co-transfected with human KOR in the presence of luciferase-based cAMP biosensor (GloSensor; Promega) to quantify KOR Gαi-mediated cAMP inhibition. After at least 16 h, the transfected cells were plated with DMEM containing 1% dialyzed FBS at a density of 15,000−20,000 cells per 40 μl per well into 96-well white clear bottom cell culture plates that were previously coated with poly-L-lysine. Following an overnight incubation at 37 °C with 5% $CO_2$, the plate medium was decanted and 20 μl per well of drug buffer (20 mM HEPES, 1x Hank's balanced salt solution (HBSS), pH 7.4) were added, followed by the addition of 10 μl of 3x drug solution that was freshly prepared in a drug buffer supplemented with 0.3% bovine serum albumin (BSA). After a 15 min incubation in the dark at room temperature, 10 μl luciferin (4 mM final concentration) that contained isoproterenol (400 nM final concentration) were added per well to induce endogenous cAMP through β adrenergic-Gs activation. Immediately after an additional 15 min incubation in the dark at room temperature, the luminescence counter (Wallac TriLux microbeta, Perkin Elmer) was utilized to measure the luminescence intensity. The measured luminescence intensity was plotted against the drug concentration, normalized to % U50,488 stimulation and then analyzed using the GraphPad Prism 8.0 software, which aligned with the dose response-stimulation "log(agonist) vs. response".

### Tango arrestin recruitment assay
To do Tango arrestin recruitment, a pcDNA3.1 vector containing human KOR-TEV cleavage site-ttA transcription factor was used. The HTLA stable cell line that expresses TEV fused-β-Arrestin2 (kindly provided by Dr. Richard Axel, Columbia Univ.). The previously established protocol[62] was used to design the KOR Tango constructs and conduct the assays. The KOR Tango construct was used to transfect HTLA cells. After at least 16 h, the transfected cells were plated with DMEM containing 1% dialyzed FBS at a density of 10,000−15,000 cells per 40 μl per well into 384-well white clear bottom cell culture plates that were previously coated with poly-L-lysine. The next day, 3x drug solutions were prepared in a drug buffer (1x HBSS), 20 mM HEPES, 0.3% BSA, pH 7.4) and 20 μl were added to each well to stimulate the cells and was incubated overnight at 37 °C with 5% $CO_2$. The following day, the cell media that contained drug solutions were gently aspirated, then 20 μl per well of BrightGlo reagent (purchased from Promega, after 1:20 dilution) were added and incubated for 20 min at room temperature in the dark. Thereafter, the luminescence counter was utilized to quantify the luminescence intensity that was plotted against the drug concentration. It was then normalized to % U50,488 stimulation and analyzed using the GraphPad Prism 8.0 software, which aligned with dose response-stimulation "log(agonist) vs. response".

### Bioluminescence resonance energy transfer (BRET) assay
To determine the interaction between human KOR and the seven individual Gα protein subunits (Gi1, Gi2, Gi3, GoA, GoB, Gz, and Ggustucin), individual Gα construct was fused with a Renilla luciferase

(Rluc), the Gγ construct was fused with a N-terminal GFP. All constructs (hKOR, Gα-Rluc, Gβ1, Gγ2-GFP2) were in pcDNA3.1 vector that have been previously designed in Olsen et al.[44]. To do BRET2-G protein activation, HEK293T cells were co-transfected using a 1:1:1:1 DNA ratio of KOR:Gα-RLuc8:Gβ:Gγ-GFP2. Likewise, to do BRET1-arrestin recruitment assays, HEK293T cells were co-transfected in a 1:5 DNA ratio with human KOR engineered to fuse with Renilla luciferase (KOR-RLuc8) at the C-terminus and individual β-arrestin subtypes (β-arrestin 1 or β-arrestin 2) fused with mVenus at the N-terminus (mVenus-βarrestin 1 or mVenus-βarrestin 2). Transfection reagent, transit 2020, was prepared in Opti-MEM at a ratio of 2 μl Transit:1 μg DNA, incubated for 40 min, then directly added dropwise to the cells. The next day, the transfected cells were plated with DMEM supplemented with 1% dialyzed FBS at a density of 40–50,000 cells per 200 μl per well into 96-well white clear bottom cell culture plates that were previously coated with poly-L-lysine. A day later, the cells were washed with 60 μL of a drug buffer (1x HBSS), 20 mM HEPES, pH 7.4) per well after gently aspirating the media, followed by the addition of 60 μL of the RLuc substrate, coelenterazine 400a (5 μM final concentration in drug buffer) or RLuc substrate, coelenterazine h (5 μM final concentration in drug buffer) per well to activate Gα protein or arrestin, respectively. Following 5 min of incubation at room temperature in the dark, 30 μL of drug (3x) were added in a drug buffer (1x HBSS), 20 mM HEPES, 0.3% BSA), pH 7.4) per well and incubated for an additional 5 min. Subsequently, Mithras LB940 multimode microplate reader was used for measuring the BRET ratio for 1 s per well by detecting the ratio of the GFP2 emission at 510 nm to Rluc emission at 395 nm for Gα protein activation, whereas the ratio of mVenus emission at 485 nm to Rluc emission at 530 nm for arrestin recruitment. Using Graphpad Prism 8 software, the BRET ratios were plotted against drug concentration, normalized, and then analyzed to detect the examined drug's potency and efficacy.

## Dose response, log(τ/$K_A$) calculation, and ligand bias quantification
Bias factor toward G protein was calculated with U50,488 as a reference agonist. In detail,

(1) Dose–response data with respect to reference ligands (G protein signaling or arrestin signaling) were fit using the Black and Leff operational model[63] (B = 10^(X*n)*10^(Logτ*n);  Y = Emax*B/[B + (10^X + 10^LogK_A)^n]) in Graphpad Prism 9.0, where Emax represents the maximum response of the system and was set to 100, $K_A$ is the functional dissociation constant for the agonist, τ is the efficacy of the agonist in the given pathway, and n is shared hill slope.

(2) The transduction coefficients (log(τ/$K_A$)) for each pathway and each ligand were calculated using the Black and Leff operational in Graphpad Prism 9.0.

(3) Using U50,488 as the full agonist reference, transduction coefficients for Gi (cAMP inhibition or BRET2) and β-Arrestin2 (Tango recruitment or BRET1) were calculated and averaged across experiments. For G protein-cAMP inhibition, $\Delta\log(\tau/K_A)_{nalfurafine/G\ pathway} = \log(\tau/K_A)_{nalfurafine} - \log(\tau/K_A)_{U50,488}$; for Tango-arresin recruitment, $\Delta\log(\tau/K_A)_{nalfurafine/arrestin\ pathway} = \log(\tau/K_A)_{nalfurafine} - \log(\tau/K_A)_{U50,488}$.

(4) The $\Delta\Delta\log(\tau/K_A)$ was calculated by subtracting the Gi transduction coefficient from the β-Arrestin2 transduction coefficient. $\Delta\Delta\log(\tau/K_A)_{nalfurafine} = \Delta\log(\tau/K_A)_{nalfurafine/G\ pathway} - \Delta\log(\tau/K_A)_{nalfurafine/arrestin\ pathway}$. Calculation of bias factors utilized the method by Kenakin et al.[64], and bias factor of nalfurafine = $10^{\Delta\Delta\log(\tau/KA)nalfurafine}$.

## Radioligand binding assay and ligand dissociation assay
Sf9 membrane fractions that expressed the crystallization construct BRIL-KOR or HEK293T membrane preparations, that transiently expressed KOR wt or KOR mutants, were used to conduct binding experiments. The standard binding buffer (50 mM Tris, 0.1 mM EDTA, 10 mM MgCl$_2$, 0.1% BSA, pH 7.4) was used to perform the binding assay in 96-well plates. The competition binding reaction was initiated by adding 50 μL of ³H-Diprenorphine (final concentration = 1 nM in 150 μL) to the assay mixture that contained 50 μL of tested ligand and 50 μL of homogenous membrane solution that prepared in standard binding buffer in 96-well plate. The concentration of ³H-Diprenorphine was 1 nM in all wells, while varying concentrations of the tested ligands (e.g., nalfurafine) ranging from −5 to −12 were added as shown in the x-axis. Binding reactions were allowed to proceed for 2 h at room temperature in the dark. After this, rapid vacuum filtration onto chilled 0.3% polyethylenimine-soaked GF/A filters followed by three quick washes with a cold washing buffer (50 mM Tris HCl, pH 7.4) was carried out to terminate the binding reaction. Radioligand activity was quantified by liquid scintillation using the MicroBeta counter. GraphPad Prism 8.0 software was employed to analyze the binding results (with or without normalization) that aligned to one-site-Ki equation.

Similar to radioligand binding assay, radioligand dissociation assay was conducted in the standard binding buffer in 96-well plates. Two radioligand ([³H]-U69,593 = 0.5 or 2.0 nM) (PerkinElmer) concentrations were used in all assays. Membranes were incubated with radioligand for at least 2 h at 37 °C in the absence or presence of Gi1 or Nb39, followed by the addition of 10 μL of 10 μM excess cold ligand (U69,593) to the 200 μL membrane suspension at designated time points to conduct dissociation assays. Time points ranged between 2 min to 2 h. 10 μM JDTic for KOR was added to detect non-specific binding. At this time, which was defined as time 0, vacuum filtration onto 0.3% polyethyleneimine pre-soaked 96-well filter mats (Perkin Elmer) using a 96-well Filtermate harvester, followed by three washes of a cold wash buffer (50 mM Tris pH 7.4) was performed to harvest the plates. Scintillation (Meltilex) cocktail sheets (Perkin Elmer) were placed on top of the dried filters and heated at 90 °C on a heating plate, and then the Wallac Trilux MicroBeta counter (PerkinElmer) was used to count the radioactivity. Dissociation data were fitted to "Dissociation – One phase exponential decay" using Graphpad Prism 8.0 software.

The receptor expression level was measured by single-point radioligand binding assays. First, KOR WT and mutants were individually expressed in HEK 293T cells for 48 h. The cells membrane was then prepared and resuspended in standard binding buffer. Total protein concentration of each membrane resuspension was measured. In 96-well plate in triplicate, the assay mixture containing 50 μL of ³H-Diprenorphine (final concentration was 1 nM or 5 nM for two independent assays), 50 μL of standard binding buffer, and 50 μL of homogenous membrane solution were subsequently incubated in the dark at room temperature for 2 h. Then, the plate was harvested by a cell harvester and radioactivity (count per minute, CPM) was recorded by MicroBeta2. For data normalization, the CPM values were divided by the total proteins used to obtain the results as shown in the y-axis "CPM ³H-diprenorphine/μg total protein" (Supplemental Fig. 8).

## System setup for molecular dynamics simulations
We performed simulations of KOR bound to nalfurafine, U50,488, and WMS-X600. The simulations were initiated using the nalfurafine-bound structure reported in this manuscript, with the nanobody removed. The S324C mutation introduced for structure determination was reverted. For the U50,688 and WMS-X600 simulations, nalfurafine was removed and the given ligand was docked into the structure using Glide SP (Schrödinger L.L.C.). For each ligand, many poses were considered, and a pose was chosen that recreated the key interactions present in other opioid structures: a salt bridge to D[3.32], occupancy of the central cavity by a bulky aliphatic group, and placement of an aromatic ring between TM2 and TM3. For each ligand, we performed ten independent simulations, each 3.0 μs in length. For each simulation, initial atom velocities were assigned randomly and independently.

The structure was aligned to the Orientations of Proteins in Membranes[65] entry for 5C1M (active μOR bound to BU72[37]), and crystal waters from 5C1M were incorporated. Prime (Schrödinger)[66] was used to model missing side chains, and to add capping groups to protein chain termini. The Crosslink Proteins tool (Schrödinger) was used to model unresolved portions of ECL2, ICL3, and ECL3. Parameters for the ligands were generated using the Paramchem webserver[67–69]. The parameters for nalfurafine were optimized to quantum calculations performed using Gaussian. Protonation states of all titratable residues were assigned at pH 7, except for $D^{2.50}$ and $D^{3.49}$, which were protonated (neutral) in all simulations, as these conserved residues are reported to be protonated in active-state GPCRs[70–73]. Histidine residues were modeled as neutral, with a hydrogen atom bound to either the delta or epsilon nitrogen depending on which tautomeric state optimized the local hydrogen-bonding network. Using Dabble[74], the prepared protein structures were inserted into a pre-equilibrated palmitoyl-oleoyl-phosphatidylcholine (POPC) bilayer, the system was solvated, and sodium and chloride ions were added to neutralize the system and to obtain a final concentration of 150 mM. The final systems comprised approximately 59,490 atoms, and system dimensions were approximately $80 \times 80 \times 90$ Å.

## Molecular dynamics simulation and analysis protocols

We used the CHARMM36m force field for proteins, the CHARMM36 force field for lipids and ions, and the TIP3P model for waters[75–77]. All simulations were performed using the Compute Unified Device Architecture (CUDA) version of particle-mesh Ewald molecular dynamics (PMEMD) in AMBER18[78] on graphics processing units (GPUs).

Systems were first minimized using three rounds of minimization, each consisting of 500 cycles of steepest descent followed by 500 cycles of conjugate gradient optimization. 10.0 and 5.0 kcal•mol$^{-1}$•Å$^{-2}$ harmonic restraints were applied to protein, lipids, and ligand for the first and second rounds of minimization, respectively. 1 kcal•mol$^{-1}$•Å$^{-2}$ harmonic restraints were applied to protein and ligand for the third round of minimization. Systems were then heated from 0 K to 100 K in the NVT ensemble over 12.5 ps and then from 100 K to 310 K in the NPT ensemble over 125 ps, using 10.0 kcal•mol$^{-1}$•Å$^{-2}$ harmonic restraints applied to protein and ligand heavy atoms. Subsequently, systems were equilibrated at 310 K and 1 bar in the NPT ensemble, with harmonic restraints on the protein and ligand non-hydrogen atoms tapered off by 1.0 kcal•mol$^{-1}$•Å$^{-2}$ starting at 5.0 kcal•mol$^{-1}$•Å$^{-2}$ in a stepwise fashion every 2 ns for 10 ns, and then by 0.1 kcal•mol$^{-1}$•Å$^{-2}$ every 2 ns for 20 ns. Production simulations were performed without restraints at 310 K and 1 bar in the NPT ensemble using the Langevin thermostat and the Monte Carlo barostat, and using a timestep of 4.0 fs with hydrogen mass repartitioning[79]. Bond lengths were constrained using the SHAKE algorithm[80]. Non-bonded interactions were cut off at 9.0 Å, and long-range electrostatic interactions were calculated using the particle-mesh Ewald (PME) method with an Ewald coefficient of approximately 0.31 Å, and 4th order B-splines. The PME grid size was chosen such that the width of a grid cell was approximately 1 Å. Trajectory frames were saved every 200 ps during the production simulations.

TM7 rotation at $S324^{7.47}$ was calculated by first aligning each simulation frame to the starting structure, which was itself aligned to the OPM entry for 5C1M using TM2, TM3, and TM5. Next, a vector approximately tangent to the helix at $S324^{7.47}$ was computed by taking the difference in XY position of the Cα atoms of $S324^{7.47}$ and $L325^{7.48}$. We report the angle between these vectors in each simulation frame and the reference structure.

## Reporting summary

Further information on research design is available in the Nature Portfolio Reporting Summary linked to this article.

## Data availability

The data that support this study are available from the corresponding authors upon request. The KOR-Nalfurafine-Nb39 data generated in this study have been deposited in the Protein Data Bank (PDB) under accession code 7YIT. The processed statistics were summarized in Supplemental Table 1. The source data generated in this study are provided as a Source data file. Source data are provided with this paper.

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

## Acknowledgements

This work was supported by National Institutes of Health grant R35GM143061 (T.C.). We would like to thank Carl-Mikael Suomivuori and Deniz Aydin for helpful discussion of simulation data. This work was supported by National Institutes of Health grant R01GM127359 (R.O.D.). This work was supported by National Institutes of Health grant F31-NS093917 (R.H.J.O.). An award of computer time was provided by the INCITE program. This research used resources of the Oak Ridge Lea-dership Computing Facility, which is a U.S. Department of Energy Office of Science User Facility supported under contract DE-AC05-00OR22725. P.P. acknowledges RSF research project No. 22-74-10098. We acknowledge the NIDA Drug Supply Program for providing the $^3$H-JDTic. We thank J. Smith and R. Fischetti and the staff of GM/CA@APS, which has been funded with Federal funds from the National Cancer Institute (ACB-12002) and the National Institute of General Medical Sciences (AGM-12006). This research used resources of the Advanced Photon Source, a U.S. Department of Energy (DOE) Office of Science User Facility, operated for the DOE Office of Science by Argonne National Laboratory (contract no. DE-AC02-06CH11357).

## Author contributions

A.E. analyzed the structure, performed ligand binding and functional assays, analyzed all data, and prepared the manuscript. J.M.P. per-formed the molecular dynamic simulation, analyzed the data, and pre-pared the manuscript. K.K. helped with diffraction data collection and processed diffraction data and structure refinement. Y.L. performed the molecular dynamic simulation, analyzed the data. P.P. helped identify thermostabilized mutations for crystallization. S.B. helped perform functional assays for transducerome analysis. B.E.K. helped collect the diffraction data. R.H.J.O. and J.D. helped with transducerome screening and optimization. F.I.C. provided nalfurafine ligand. V.C. helped with identify thermostabilized mutations for crystallization and manuscript discussion. B.W. provided the ligand WMS-X600. R.O.D. supervised the molecular dynamic simulation, helped with data discussion, and manuscript preparation. T.C. designed the experiments, was respon-sible for the overall project strategy and management and prepared the manuscript.

## Competing interests

The authors declare no competing interests.
