## [Peer Review File · Nature Communications]

Molecular mechanism of biased signaling at the kappa opioid receptorReviewers' Comments:

Reviewer #1:

Remarks to the Author:

This manuscript from Daibani etc. reports mechanistic studies on the biased signaling properties of two KOR agonists, nalfurafine as a Gi-biased agonist and WMS-X600 as a b-arrestin-biased agonist. The biased signaling of opioid receptors has attracted much research interest due to the therapeutic potential of biased opioid agonists as the next-generation analgesics. In this paper, the authors determined a crystal structure of KOR with nalfurafine as the first KOR-targeting drug, performed MD simulations on KOR with three agonists showing different bias profiles, and performed extensive cell-based signaling assays to identify critical structural features of KOR associated with ligand bias. This work provided novel molecular insights into how KOR agonists induce different conformations of KOR to result in arrestin-dependent and G protein-dependent signaling events. The senior authors are top experts in their fields of study. The quality and impact of the study are high. -Cheng

Several minor issues:

1. For opioid receptors, it has been shown that the agonist-binding and the Gi-coupling events are loosely coupled. It is interesting to see that Nb39 can cause a much higher increase of nalfurafine affinity compared to that caused by Gi, meaning a stronger allosteric effect of Nb39 on agonist binding. Does that mean Nb39 can stabilize a more 'active' conformation of KOR? Is there any structural explanation for this, e.g. a more open conformation of the intracellular cavity stabilized by Nb39?
2. Line 158, the authors stated that the mutation of W6.48 to leucine had minimal effect on G protein activation, yet they further suggested that W6.48 plays a key role in mediating ligand efficacy in Gi signaling. This needs a more detailed explanation. If the steric distortion of W6.48 caused by the bulky cyclopropylmethyl group of nalfurafine is an important event in KOR activation, would a leucine residue at 6.48 with a much smaller side chain somehow weaken such steric effect and result in 'less' active KOR?
3. To be consistent and to help readers to get where the residues discussed, it may be better to show BW numbering for all residues of KOR mentioned in the paper.
4. Line 278, nalfurafine can cause the receptor to sample the occluded state more often. How does that lead to G protein-biased signaling? Is this occluded state more suitable for Gi coupling than for arrestin binding? It doesn't seem so based on Figure 4A
5. The authors used the TRUPATH assay to extensively characterize the bias properties of four agonists and identify residues that play different roles in activating different signal transducers. The results clearly provided valuable information for future studies on the biased signaling of opioid receptors. However, all signaling assays were performed in HEK293 cells with the recombinantly expressed receptor or G proteins. Do the results in Figure 5 reflect the real bias properties of ligands in physiological conditions considering other factors like availability of different Gi protein subtypes in the CNS? This may be beyond the scope of this study. However, it will be helpful if the authors can comment on it and discuss potential limitations of the study in the Discussion section.
6. This is related to 5. Nalfurafine is clearly a G protein biased agonist, but it is not a complete G protein biased agonist that induces minimal b-arrestin recruitment. The EC50 of nalfurafine is still in the single-digit nanomolar range. When used as a drug, it is likely that nalfurafine can still induce strong b-arrestin-dependent signaling events in vivo. If that is the case, how does the G protein-biased property of nalfurafine translate into the minimal side effects of dysphoria and hallucination at therapeutic doses?

Reviewer #2:

Remarks to the Author:

The manuscript by El Daibani et al. describes a new kappa opioid receptor (KOR) structure bound with a G-protein biased agonist nalfurafine. Based on this structure, molecule dynamics (MD) simulations of nalfurafine, WMS-X600, and U50,488 bound KOR models identified three conformational states that may favor arrestin or G protein signaling. Then the transducerome profiling was carried out for three ligands against seven G proteins and two arrestins.

The main concern starts from the crystal structure of the KOR/nalfurafine-Nb39 complex. As the authors described in lines 111-117 on page 6, "This is likely due to the application of the same nanobody (Nb39) during the co-crystallization as it recognizes a specific active state...", it is not clear whether the Nb39 stabilized conformation is closer to G-protein or arrestin biased one. Thus, the structure itself may not really tell much about the "insights into biased signaling", if not misleading. In addition, recent report on the D1 structures bound with or without another nanobody Nb35 (PMID 35687690) suggested "that Nb35 binding allosterically influences the conformation of the agonist-binding pocket". Thus, Nb39's impact on the receptor conformation and in particular the agonist-binding pocket is a concern. As the structure is the starting point for the MD simulations, while the real nalfurafine favored KOR conformation is supposed to be more or less different from that stabilized by Nb39, it is important to demonstrate that the MD simulations were extensive enough so they had overcome the energy barriers introduced by Nb39.

If the crystal structure serves a purpose of providing a relatively accurate binding pose of nalfurafine, and it takes a crystal structure to do that, then the question is how reliable the binding poses of U50,488 and WMS-X600 acquired by docking are. The criteria in choosing the docking poses described in lines 627-630 on page 25, "a salt bridge to D3.32, occupancy of the central cavity by a bulky aliphatic group, and placement of an aromatic ring between TM2 and TM3" sound too vague for a high-resolution question of the "insights into biased signaling" – how much the binding poses of each ligand have diverged in the MD simulations from their starting points needs to be characterized. Because the crystal structure has limited help in answering the "insights" question, the bar for the simulation work has to be higher: An analysis should be carried out to show that the simulations are converged, e.g., the populations of each state in Fig. 4B do not further change with additional simulations.

It is difficult to put the highest functional potency shown at Gz by all three agonists (Fig 5A) in the context of the "insights" based on the analysis of the crystal structure and MD simulations. For example, does the conformational ensemble favored by the arrestin-biased WMS-X600 also favor Gz signaling, while coupling to Gz is likely closer to Gi than to any of arrestins?

Reviewer #3:

Remarks to the Author:

There is also concern with the assertion that there are "single" confirmations that will determine bias in one direction over another- the title, the abstract etc. For example, we know that mutations in G protein or arrestin binding domains of GPCRs will confer bias- that may not be the same as those described here. Further, agonists that bind in allosteric sites may also confer bias that may differ from that described here. This is a suggestion to modify the tone to indicate "a" rather than "the" molecular basis for biased signaling at KOR. This is warranted given the very little degree of bias observed with the chosen ligands.

The mutational analyses are extensive but I can find no description of the generation of the cell lines. Are the expression levels equivalent? Normalization to the effect of U69 allows for comparison of EC50 values, but did the mutations effect the overall efficacy of U69? Inclusion of the fold stimulation observed in the supplemental would be helpful. It is possible to maintain potency and loose efficacy-

did this happen?

Writing style- the writing is dense and there is a great deal of discussion and description/introduction included in the results section making it difficult to grasp the actual results. The manuscript would be improved if the data were presented with interpretation in the results section, without as much discussion (save for discussion).

This is particularly an issue with the presentation of the "transcriptome" profiling section- while the nature of the G proteins are described in detail- what is the conclusion of this profiling? Is it just a collection of data? Again, analysis of the degree of bias should be included for this set, as biased agonism is not just a measure of propensity to signal to G protein vs. arrestin (as they note)- it can also be a difference between the ability to signal to Gi vs Gz. If this is the case, how does this alter the conclusions of the manuscript? This result section should discuss the results and not describe the action of each G protein. How does Gz vs. Barrestin recruitment fair regarding bias? Is it conserved? Does the argument hold true then that these modeled confirmations determine "the" pose for biased agonism?

Overall, the result section needs to describe the experiments herein and the discussion can be used to put the results observed herein in context with prior studies. As it stands, upon reading the paper, I'm left digging through the figures trying to determine what new idea is here. I think that the work is extensive- but there remains an opportunity to present the findings in a more focused way.

REVIEWER COMMENTS

Reviewer #1 (Remarks to the Author):

This manuscript from Daibani etc. reports mechanistic studies on the biased signaling properties of two KOR agonists, nalfurafine as a Gi-biased agonist and WMS-X600 as a b-arrestin-biased agonist. The biased signaling of opioid receptors has attracted much research interest due to the therapeutic potential of biased opioid agonists as the next-generation analgesics. In this paper, the authors determined a crystal structure of KOR with nalfurafine as the first KOR-targeting drug, performed MD simulations on KOR with three agonists showing different bias profiles, and performed extensive cell-based signaling assays to identify critical structural features of KOR associated with ligand bias. This work provided novel molecular insights into how KOR agonists induce different conformations of KOR to result in arrestin-dependent and G protein-dependent signaling events. The senior authors are top experts in their fields of study. The quality and impact of the study are high. -Cheng

We thank the reviewer for evaluating of our manuscript and providing us with the positive comments regarding both the quality and impact of this study.

Several minor issues:

1. For opioid receptors, it has been shown that the agonist-binding and the Gi-coupling events are loosely coupled. It is interesting to see that Nb39 can cause a much higher increase of nalfurafine affinity compared to that caused by Gi, meaning a stronger allosteric effect of Nb39 on agonist binding. Does that mean Nb39 can stabilize a more 'active' conformation of KOR? Is there any structural explanation for this, e.g. a more open conformation of the intracellular cavity stabilized by Nb39?

Our pharmacological and structural evidence supports that the Nb39 stabilizes a different conformation from the Gi-bound state, although it is difficult to conclude if it is a more active conformation. We compared the TM6 in Nb39-bound KOR with Gi-bound MOR (Fig. 1F in the main figures) and we observed a 2 Å dislocation between these two structures. Further support comes from the comparisons of MOR-BU72-Nb39 and MOR-DAMGO-Gi1 showing that there is a 3 Å dislocation of TM6 (Koehl A et al., Nature, 2018). As the Nb39 forms unique interactions with the intracellular KOR, particularly Intracellular Loop 3 (ICL3), which is different from the interactions with Gi1 (Koehl A et al., Nature, 2018), we believe that these interactions could be a reason for a more open conformation induced by Nb39.

2. Line 158, the authors stated that the mutation of W6.48 to leucine had minimal effect on G protein activation, yet they further suggested that W6.48 plays a key role in mediating ligand efficacy in Gi signaling. This needs a more detailed explanation. If the steric distortion of W6.48 caused by the bulky cyclopropylmethyl group of nalfurafine is an important event in KOR activation, would a leucine residue at 6.48 with a much smaller side chain somehow weaken such steric effect and result in 'less' active KOR?

Thank you for pointing this out. W6.48 is known as the toggle switch involved in class A GPCR activation. Relevant biased signaling studies in MOR found that different agonist binding poses with respect to W6.48 might be the initial trigger of the different signaling outcomes (Cong et al., Mol Cell 2021). We have made the new W6.48A mutation and tested it with U50,488, nalfurafine, and WMS-X600 (Fig 1). Nalfurafine mediated arrestin recruitment was significantly impaired, while the G protein signaling was slightly affected. Importantly, the W6.48A has different effects on U50,488 and WMS-X600 from nalfurafine, which we found that the mutational effect of this residue is ligand-specific and pathway-specific which is consistent with the biased signaling studies in MOR. We have included the result for W6.48A mutation in Supplemental Figure. 3 and 5 in the manuscript.

Fig1. Effects of W6.48A on ligand-dependent G protein activation and arrestin recruitment. For each ligand (top, nalfurafine; middle, WMS-X600; bottom, U50,488), G protein mediated cAMP inhibition and arrestin-mediated recruitment were measured. The dash line shows an estimate of EC50 in wild type or W6.48A mutant.

3. To be consistent and to help readers to get where are the residues discussed, it may be better to show BW numbering for all residues of KOR mentioned in the paper.

Thank you for the advice. We have included the BW numbering for all KOR residues discussed in the manuscript including Figures and Tables.

4. Line 278, nalfurafine can cause the receptor to sample the occluded state more often. How does that lead to G protein-biased signaling? Is this occluded state more suitable for Gi coupling than for arrestin binding? It doesn't seem so based on Figure 4A

Yes, we believe that the occluded state is more suitable for Gi coupling than for arrestin coupling. In the occluded state, D334^{8.47} interacts closely with T94^{2.39}. Arrestin-GPCR complex structures have revealed that a key interaction for arrestin binding is hydrogen bonding of an aspartate on the arrestin finger loop to T94^{2.39}. In the occluded state, D334^{8.47} would compete for this interaction site. In Supplementary Figure 9, we compare the complementarity of the occluded state with beta-arrestin-1 and Gi. We have also updated the text to clarify this.

5. The authors used the TRUPATH assay to extensively characterize the bias properties of four agonists and identify residues that play different roles in activating different signal transducers. The results clearly provided valuable information for future studies on the biased signaling of opioid receptors. However, all signaling assays were performed in HEK293 cells with the recombinantly expressed receptor or G proteins. Do the results in Figure 5 reflect the real bias properties of ligands in physiological conditions considering other factors like availability of different Gi protein subtypes in the CNS? This may be beyond the scope of this study. However, it will be helpful if the authors can comment on it and discuss potential limitations of the study in the Discussion section.

We agree with the reviewer that this is worth exploring in the physiological conditions in the future studies. The primary goal of this study is to examine and compare the ligand-specific transducer coupling. We used a recombinant and overexpressed system, and compared all G proteins and arrestins under the similar settings. Two related studies from Ho et al (Ho et al., *Sci Signal* 2018) and Schmid et al (Schmid et al., *Cell* 2017) have shown that KOR or MOR G protein biased signaling is maintained in neurons and in animal models. It is still possible that neurons may have different G protein subtypes preference and expression levels, we have expanded the discussion section stating that the results are obtained from the cell lines and cannot be extended to the situation in the neurons without further experiments in the endogenous systems.

6. This is related to 5. Nalfurafine is clearly a G protein biased agonist, but it is not a complete G protein biased agonist that induces minimal b-arrestin recruitment. The EC50 of nalfurafine is still in the single-digit nanomolar range. When used as a drug, it is likely that nalfurafine can still induce strong b-arrestin-dependent signaling events in vivo. If that is the case, how does the G protein-biased property of nalfurafine translate into the minimal side effects of dysphoria and hallucination at therapeutic doses?

The reviewer raises an important question about how the in vitro results on biased agonists translate into in vivo effects. Dr. Liu and his colleagues (Liu et al., *Neuropsychopharmacology*, 2018) have used a phosphoproteomic approach to profile

the nalfurafine and other reference ligands in vivo (animal model) and have found that nalfurafine engaged different signaling pathways from other KOR agonists. For example, U50,488, but not nalfurafine, can activate the mammalian target of rapamycin (mTOR) pathway, which was later confirmed as a potential mediator for KOR-mediated dysphoria. However, whether the different signaling pathways are a result of the biased signaling of nalfurafine remains to be investigated. Another observation that may correlate with nalfurafine's mild bias is the small therapeutic window of nalfurafine. At a low dose (2.5-20 $\mu\text{g}/\text{Kg}$), nalfurafine is selective at KOR and does not produce typical side effects (dysphoria, sedation, hallucinations). At high dose (20-50 $\mu\text{g}/\text{Kg}$), nalfurafine starts causing motor incoordination, sedation, and aversion effects (Liu et al., *Neuropsychopharmacology*, 2018; Abraham et al., *Neuropsychopharmacology*, 2018).

Reviewer #2 (Remarks to the Author):

The manuscript by El Daibani et al. describes a new kappa opioid receptor (KOR) structure bound with a G-protein biased agonist nalfurafine. Based on this structure, molecule dynamics (MD) simulations of nalfurafine, WMS-X600, and U50,488 bound KOR models identified three conformational states that may favor arrestin or G protein signaling. Then the transducerome profiling was carried out for three ligands against seven G proteins and two arrestins.

The main concern starts from the crystal structure of the KOR/nalfurafine-Nb39 complex. As the authors described in lines 111-117 on page 6, "This is likely due to the application of the same nanobody (Nb39) during the co-crystallization as it recognizes a specific active state...", it is not clear whether the Nb39 stabilized conformation is closer to G-protein or arrestin biased one. Thus, the structure itself may not really tell much about the "insights into biased signaling", if not misleading.

We agree with the reviewer that the nanobody-bound state may not fully represent the G protein or arrestin-bound state. As a result of no available arrestin-bound KOR, it is difficult to conclude whether the Nb39 stabilized conformation is closer to G protein or arrestin coupled state. Thus, we compared the structures of KOR-nalfurafine-Nb39, MOR-DAMGO-Gi1 (PDB 6DDF), and NTS1R-neurotensin-beta-arrestin1 (PDB 6UP7), respectively. As shown in Fig. 2A, the Nb39 stabilized KOR conformation is different from either Gi or beta-arrestin1 coupled structures (based on TM6 movement), but shows a closer conformation to MOR-Gi1. Furthermore, we identified that nalfurafine is more prone to inducing a occluded KOR state using MD simulations. In the occluded state, D334^{8.47} interacts closely with T94^{2.39}. Arrestin-GPCR complex structures have revealed that a key interaction for arrestin binding is hydrogen bonding of an aspartate on the arrestin finger loop to T94^{2.39}. In the occluded state, D334^{8.47} would compete for this interaction site (Fig. 2B). This is largely consistent with pharmacological studies from other GPCRs showing that the nanobody acts as a G protein mimetic in the stabilization of active-state receptors (Rasmussen et al., *Nature* 2011; Staus et al., *Nature* 2016). It is worth pointing out that Nb39-bound KOR or MOR both displays conformational differences from the Gi-bound MOR (Fig. 1F in this manuscript and Huang et al., *Nature* 2018). While the Nb39 stabilized structure provides valuable information how nalfurafine binds to KOR, it also serves as a template (as we provide

rationale below) for the MD simulation study. We have also included this to the Supplemental Figure 9 in the manuscript.

Fig 2. The Nb39 stabilizes a conformation more similar to Gi coupled state compared to arrestin. A. The comparison of TM6 outward movement between, KOR-Nb39, MOR-Gi1, and NTS1R- β arrestin1. B. Nalfurafine induces an occluded state that is more suitable for Gi coupling than for arrestin coupling.

In addition, recent report on the D1 structures bound with or without another nanobody Nb35 (PMID 35687690) suggested “that Nb35 binding allosterically influences the conformation of the agonist-binding pocket”. Thus, Nb39’s impact on the receptor conformation and in particular the agonist-binding pocket is a concern. As the structure is the starting point for the MD simulations, while the real nalfurafine favored KOR conformation is supposed to be more or less different from that stabilized by Nb39, it is important to demonstrate that the MD simulations were extensive enough so they had overcome the energy barriers introduced by Nb39.

In the below two responses, we have provided information about the average pose of nalfurafine in simulation (as well as for U50,488 and WMS-X600) and evidence that the simulations have converged. The deviation between the average pose in MD and the

crystallographic pose is of a similar magnitude to the difference between the two D1 structures cited by the reviewer. We note that the minor deviation between the ligand pose in crystallographic condition and in simulation does not suggest that either is “incorrect” as the conditions are very different (the temperature differs, in the simulations there is no intracellular binding partner, etc). Moreover, our simulations often visit conformational states significantly different than the crystallographic conformation (the alternative and occluded states) supporting that the simulations are not trapped near the crystallographic conformation.

If the crystal structure serves a purpose of providing a relatively accurate binding pose of nalfurafine, and it takes a crystal structure to do that, then the question is how reliable the binding poses of U50,488 and WMS-X600 acquired by docking are. The criteria in choosing the docking poses described in lines 627-630 on page 25, “a salt bridge to D3.32, occupancy of the central cavity by a bulky aliphatic group, and placement of an aromatic ring between TM2 and TM3” sound too vague for a high-resolution question of the “insights into biased signaling” – how much the binding poses of each ligand have diverged in the MD simulations from their starting points needs to be characterized. It would of course always be helpful to have a crystallographic pose from which to initiate all the simulations—making the nalfurafine-bound crystal structure valuable—however, it isn’t feasible for us to solve additional structures. Apart from serving as a starting point for simulations, the nalfurafine-bound structure provided insight into the conformational changes associated with the movements of TM6 and TM7, along with unique interactions of nalfurafine in the orthosteric binding pocket.

The docked poses for U50,488 and WMS-X600 were simply used as a starting point for simulations. We believe that the listed criteria, along with the energy evaluation by the docking software are sufficient to get us close enough for the simulations to converge to the biologically relevant pose. Indeed, the average pose of U50,488 and WMS-X600 during the simulations differs from the docked poses, but the features we listed remain present (Fig 3). We’ve added this to Supplementary Figure 4B comparing the docked poses (and crystallographic pose for nalfurafine) to the average ligand pose in the simulations. It is also worth pointing out that our simulation data agrees with the experimental validation in mutagenesis studies, which supports the convergence of the structural observation and MD simulations.

Fig 3. Average ligand and binding pocket conformations in MD simulations. In cyan, the simulation frame where the ligand and binding site are most similar (lowest RMSD) to their average coordinates across all MD simulations with the indicated ligand bound. The most similar simulation frame to the average structure is shown as opposed to the average structure itself because the average structure contains non-physical molecular geometries. In magenta: the crystallographic pose of Nalfurafine and the docked poses for U50,488 and WMS-X600, which were used to initiate the simulations.

Because the crystal structure has limited help in answering the “insights” question, the bar for the simulation work has to be higher: An analysis should be carried out to show that the simulations are converged, e.g., the populations of each state in Fig. 4B do not further change with additional simulations.

We’ve carried out two analyses to ensure that our simulations are converged.

1. We verified that our simulations are of sufficient length using the Lyman-Zuckerman method (Lyman, E. & Zuckerman, D. M. Ensemble-based convergence analysis of biomolecular trajectories. *Biophys J* 91, 164-172, doi:10.1529/biophysj.106.082941 (2006).). We split each simulation into two halves, analyzed each independently, and found that the results don’t change between the earlier and later half (Fig 4). (Note that our simulations are 3.0 μ s, and we drop the first 1.5 μ s to allow the simulations to converge. Here by two halves, we are referring to 1.5–2.25 μ s and 2.25–3.0 μ s of each simulation). We’ve added this analysis to Supplementary Figure 6A.
2. We verified that we had performed a sufficient number of simulations under each condition using a statistical analysis of state occupancies, treating each replicate as one data point. This analysis was present in our initial submission, we are including this for completeness.

Fig 4. MD simulation convergence analysis. Each of the left and right columns are analogous to Figure 4 in the manuscript. The left column uses only the simulation frames from 1.5 μ s to 2.25 μ s and the right column uses only simulation frames from 2.25 μ s to 3.0 μ s. Note that our simulations are 3.0 μ s, and we drop the first 1.5 μ s to allow the simulations to converge, so this corresponds to the first and second half of the frames shown in Figure 4 in the manuscript. The similar populations of each state in either time split suggests that the populations would not change significantly if the simulations were extended.

It is difficult to put the highest functional potency shown at Gz by all three agonists (Fig 5A) in the context of the “insights” based on the analysis of the crystal structure and MD simulations. For example, does the conformational ensemble favored by the arrestin-biased WMS-X600 also favor Gz signaling, while coupling to Gz is likely closer to Gi than to any of arrestins?

The crystal structure of KOR-nalfurafine, the conformational analysis, and the MD simulations of three KOR ligands that showed different conformations, particularly in the intracellular region, suggested that they may have different transducer coupling

preferences. Yes, all three KOR agonists show the highest potency at Gz but with different potencies (the potency for nalfurafine is 10 times higher than WMS-X600), while arrestin1 and 2 recruitments for WMS-X600 are more efficacious as compared to nalfurafine. We agree that we need to understand the coupling to Gz vs. arrestin with different KOR ligands that have distinct biased preferences. When we re-calculated the bias factor between Gz vs. β -arrestin2 pathway using U50,488 (Gz and β -arrestin2 pathways) as a reference. For nalfurafine, the bias factor towards Gz protein is 4.8; for WMS-X600, 8.6 towards arrestin pathway. This suggests that the bias profile is maintained between the Gz and β -arrestin2. Understanding the molecular basis for Gz coupling that we present in this study is highlighting the requirement for solving the structures of KOR-Gz complex and KOR-arrestin complex, which is beyond the scope of the current work.

Reviewer #3 (Remarks to the Author):

The study by Daibani et al., compares different conformations of the kappa opioid receptor when bound to agonists with different pharmacological properties. The authors propose that nalfurafine is a G protein biased agonist and WMS-X600 is a β -arrestin2 biased KOR agonist. U50488 is considered the reference agonist. The manuscript lacks analysis to determine measures of functional selectivity/bias- (no parameters or models are defined, only single digit numbers are presented in the text- is a bias factor (not sure of the derivation) of 6 considered biased?). There seems to be enough pharmacological data available that quantitative analysis could be applied with statistical confidence since this manuscript implies that it has defined structural elements that confer biased agonism.

We thank the reviewer for pointing out this methodological issue. We have now included the step-by-step details that explain how we measure and quantify the bias factor of nalfurafine and WMS-X600. Using U50,488 as a reference agonist, the $\Delta\Delta\text{Log } \tau/K_A$ and bias factor ($10^{\Delta\Delta\text{Log } \tau/K_A(\text{G protein-arrestin})}$) were calculated. The $\Delta\Delta\text{Log } \tau/K_A$ (95% confidence interval) is 0.78 (0.65 to 0.91) for nalfurafine and 1 (0.81-1.19) for WMS-X600. This has also been added to the Figure legends. The evaluation of bias factor is assay dependent. For example, nalfurafine has been found to have a bias factor of 200 toward G protein signaling when measuring ERK activation (G protein-mediated) versus p38 activation (arrestin-mediated); in other assays, nalfurafine has a bias factor of 4.5 toward G protein signaling (Cao et al., ACS Chem Neuro 2020) when using cAMP inhibition and Tango arrestin assays. In our work, we used two independent assays to measure and validate the bias factor of nalfurafine and WMS-X600. We obtained the bias factor of 6 and 10 for nalfurafine and WMS-X600, respectively, which displays 95% CI and are considered biased agonists.

There is also concern with the assertion that there are “single” confirmations that will determine bias in one direction over another- the title, the abstract etc. For example, we know that mutations in G protein or arrestin binding domains of GPCRs will confer bias- that may not be the same as those described here. Further, agonists that bind in allosteric sites may also confer bias that may differ from that described here. This is a suggestion to modify the tone to indicate “a” rather than “the” molecular basis for biased

signaling at KOR. This is warranted given the very little degree of bias observed with the chosen ligands.

We agree with the reviewer that the direction of downstream signaling is likely attributed to more complicated and dynamic conformational states of the receptor. We thus modify our descriptions to make it more neutral in the abstract. We have also described the limitations of our study in the discussion section: we provide structural and MD simulation support that indicate pharmacologically different ligands can induce different conformational states, which may contribute to the preference of transducer coupling.

The mutational analyses are extensive but I can find no description of the generation of the cell lines. Are the expression levels equivalent? Normalization to the effect of U69 allows for comparison of EC50 values, but did the mutations effect the overall efficacy of U69? Inclusion of the fold stimulation observed in the supplemental would be helpful. It is possible to maintain potency and loose efficacy- did this happen?

We have now included the cell line and constructs information for the mutational studies in the Methods. The expression level of mutants has been measured by saturation binding using cell membranes that transiently express wild type and mutated KOR (Supplemental Figure 8).

Most of the mutations examined here similarly affect the efficacy of reference U50,488 and nalfurafine. This is why we only included the EC50 values in the supplemental tables initially. To provide a complete dataset, we have now included both the fold change in the potency of nalfurafine as compared to the wild type, and the Efficacy \pm SEM in the supplemental tables (Supplemental Table S4 and S5). We did not include the data for a few other mutations that completely abolish the function (both potency and efficacy) of U50,488 or nalfurafine after confirming that their interfering with ligand binding or receptor expression.

Writing style- the writing is dense and there is a great deal of discussion and description/introduction included in the results section making it difficult to grasp the actual results. The manuscript would be improved if the data were presented with interpretation in the results section, without as much discussion (save for discussion).

We thank the reviewer for the advice. We have modified the Results and Discussion sections as shown by the red-colored labeled: 1) Results section will mainly present the experimental results, analysis, and conclusions; 2) Discussion section will discuss how our results add, support, or refute the current research, as well as the limitation of our research.

This is particularly an issue with the presentation of the “transcriptome” profiling section- while the nature of the G proteins are described in detail- what is the conclusion of this profiling? Is it just a collection of data? Again, analysis of the degree of bias should be included for this set, as biased agonism is not just a measure of propensity to signal to G protein vs. arrestin (as they note)- it can also be a difference between the ability to signal to Gi vs Gz. If this is the case, how does this alter the conclusions of the manuscript? This result section should discuss the results and not describe the action of each G protein. How does Gz vs. Barrestin recruitment fair regarding bias? Is it

conserved? Does the argument hold true then that these modeled confirmations determine “the” pose for biased agonism?

We thank the reviewer for pointing out these issues. The transducerome screening results, as well as complimentary data obtained from the structural and MD simulations, provide support that ligand-specific conformation may contribute to preference of the transducer coupling. We added more information about how residues differentially affect the G protein and arrestin pathways (some of these were not captured by the structure or MD simulation).

As the reviewer suggested, we re-calculated the bias factor between Gz vs. β -arrestin2 pathway using U50,488 (Gz and β -arrestin2 pathways) as a reference. For nalfurafine, the bias factor towards G protein is 4.8; for WMS-X600, 8.6 towards arrestin pathway. This suggests that the bias profile is maintained between the Gz and β -arrestin2. This is consistent with the three agonists (U50,488, Nalfurafine, and WMS-X600) display the similar patterns in the potency of activating individual G protein subtypes (Gz>GoB>GoA>Gi1>Gi2>Gi3). Regarding the Gustducin subtype, however, this bias is no longer conserved, which suggests that the Gustducin is likely not a canonical signal transducer of KOR.

Overall, the result section needs to describe the experiments herein and the discussion can be used to put the results observed herein in context with prior studies. As it stands, upon reading the paper, I’m left digging through the figures trying to determine what new idea is here. I think that the work is extensive- but there remains an opportunity to present the findings in a more focused way.

As kindly advised by the reviewer, we have revised the Results and Discussion sections as shown by the red-colored text.

We also included the cell line and constructs information, and the method for calculating bias factor in the Methods.

Reviewers' Comments:

Reviewer #1:

Remarks to the Author:

The authors have addressed all my concerns.

Reviewer #2:

Remarks to the Author:

I appreciate the authors' efforts in addressing my comments. However, the fundamental issue remains unsolved. The crystal structure of KOR, which appears to be a highlight of the manuscript, does not help in addressing the "insights" question asked.

Reviewer #3:

Remarks to the Author:

The authors have responded to prior critiques yet the writing could be clearer.

Minor points should be addressed:

In many cases, the graphs are labeled by default Prism, underscores are in place where they do not need to be- graph titles or Y axes could be more descriptive.

The authors are also encouraged to triple check the extensive tables- it is easy to make an error in handling so much data (there is a 1 missing from the KOR WT FC in S4 table).

The order of the supplemental tables does not go in the order of the data in the main text- some tables are labeled figures and some are labeled tables- there are so many figures and so much data per figure -keeping the progression in order will help the readers.

Figure S8- the diprenorphine binding here- is this specific binding? It isn't clear from the legend or the method. According to the response- this bar chart is the result of saturation binding- but it is hard to tell from the legend or the label.

For the radioligand binding methods-

The competition binding

641 reaction was initiated by adding 50 μ L of 3x drug solution in the presence of 3H-Diprenorphine 642 (final 1 μ M)

What is the final concentration of 1 μ M referring to- isn't this a curve with varying concentrations?
What was the concentration of 3H-dip?

REVIEWERS' COMMENTS

Reviewer #1 (Remarks to the Author):

The authors have addressed all my concerns.

Response: We would like to thank the reviewer for the output and constructive criticism.

Reviewer #2 (Remarks to the Author):

I appreciate the authors' efforts in addressing my comments. However, the fundamental issue remains unsolved. The crystal structure of KOR, which appears to be a highlight of the manuscript, does not help in addressing the “insights” question asked.

Response: We thank the reviewer for the time and constructive criticism. The structure does have limitations to address the question wanted to answer, which is also the reason we sought molecular dynamic (MD) simulations and functional assays for complementary support. We have further modified the main text to emphasize that the conclusions of this work are based on the combination studies from structural biology, pharmacology, and MD simulations. We have also followed your suggestions and the reviewer's comments, and change the title to “Molecular mechanism of biased signaling at the kappa opioid receptor”.

Reviewer #3 (Remarks to the Author):

The authors have responded to prior critiques yet the writing could be clearer.

Minor points should be addressed:

In many cases, the graphs are labeled by default Prism, underscores are in place where they do not need to be- graph titles or Y axes could be more descriptive.

Response: We do agree with reviewer that the labelling of the Prism graphs needs to be revised. Accordingly, we have modified the titles of the Prism graph and added a more descriptive label to the Y-axis.

The authors are also encouraged to triple check the extensive tables- it is easy to make an error in handling so much data (there is a 1 missing from the KOR WT FC in S4 table).

Response: We thank the reviewer for noticing that. The missing value for the FC has now been added to the Table S10. We have also checked the data in all tables provided in this manuscript.

The order of the supplemental tables does not go in the order of the data in the main text- some tables are labeled figures and some are labeled tables- there are so many figures and so much data per figure -keeping the progression in order will help the readers.

Response: As kindly pointed out, the order of the supplemental tables has been changed in the manuscript to make it easier for the reader to follow the paper.

Figure S8- the diprenorphine binding here- is this specific binding? It isn't clear from the legend or the method. According to the response- this bar chart is the result of saturation binding- but it is hard to tell from the legend or the label.

Response: We do thank the reviewer for bringing up this point.

The receptor expression level was measured by single-point radioligand binding assays. First, KOR WT and mutants were individually expressed in HEK 293T cells for 48h. The cells membrane was then prepared and resuspended in standard binding buffer. Total protein concentration of each membrane resuspension was measured. In 96-well plate in triplicate, the assay mixture containing 50 μ L of 3 H-Diprenorphine (final concentration was 1 nM or 5 nM for two independent assays), 50 μ L of standard binding buffer, and 50 μ L of homogenous membrane solution were subsequently incubated in the dark at room temperature for 2h. Then, the plate was harvested by a cell harvester and radioactivity (count per minute, CPM) was recorded by MicroBeta2. For data normalization, the CPM values were divided by the total proteins used to obtain the results as shown in the y-axis "CPM 3 H-diprenorphine / μ g total protein" (Figure S8). We have now added this information in the Figure S8 and the Methods section.

For the radioligand binding methods-

The competition binding

641 reaction was initiated by adding 50 μ L of 3x drug solution in the presence of 3 H-Diprenorphine 642 (final 1 μ M)

What is the final concentration of 1 μ M referring to- isn't this a curve with varying concentrations? What was the concentration of 3 H-dip?

Response: We thank the reviewer for noticing this error. The final concentration of 3 H-Diprenorphine is 1 nM instead of 1 μ M in the presence of various concentrations of the examined ligand, which is nalfurafine. Briefly, the competition binding assay was performed in a 96-well plate set up by row. In each row, the wells contained 50 μ L of 3 H-Diprenorphine, 50 μ L of any tested ligands, and 50 μ L of membrane solution. The concentration of 3 H-Diprenorphine is 1nM in all the wells, whereas the tested ligands (e.g., nalfurafine) were at varying concentrations ranging from -5 to -12 as shown in the x-axis.

To make this clear, we have revised the (Radioligand binding assay and ligand dissociation assay) in the Methods section.